# The insulin/IGF signaling cascade modulates SUMOylation to regulate aging and proteostasis in *Caenorhabditis elegans*

Lorna Moll[1†], Noa Roitenberg[1†], Michal Bejerano-Sagie[1], Hana Boocholez[1], Filipa Carvalhal Marques[1], Yuli Volovik[1], Tayir Elami[1], Atif Ahmed Siddiqui[1], Danielle Grushko[1], Adi Biram[2], Bar Lampert[3], Hana Achache[3], Tommer Ravid[2], Yonatan B Tzur[3], Ehud Cohen[1]*

[1]Department of Biochemistry and Molecular Biology, Institute for Medical Research Israel-Canada, The Hebrew University School of Medicine, Jerusalem, Israel; [2]Departments of Biological Chemistry, The Alexander Silberman Institute of Life Sciences, The Hebrew University of Jerusalem, Jerusalem, Israel; [3]Departments of Genetics, The Alexander Silberman Institute of Life Sciences, The Hebrew University of Jerusalem, Jerusalem, Israel

**Abstract** Although aging-regulating pathways were discovered a few decades ago, it is not entirely clear how their activities are orchestrated, to govern lifespan and proteostasis at the organismal level. Here, we utilized the nematode *Caenorhabditis elegans* to examine whether the alteration of aging, by reducing the activity of the Insulin/IGF signaling (IIS) cascade, affects protein SUMOylation. We found that IIS activity promotes the SUMOylation of the germline protein, CAR-1, thereby shortening lifespan and impairing proteostasis. In contrast, the expression of mutated CAR-1, that cannot be SUMOylated at residue 185, extends lifespan and enhances proteostasis. A mechanistic analysis indicated that CAR-1 mediates its aging-altering functions, at least partially, through the notch-like receptor *glp-1*. Our findings unveil a novel regulatory axis in which SUMOylation is utilized to integrate the aging-controlling functions of the IIS and of the germline and provide new insights into the roles of SUMOylation in the regulation of organismal aging.
DOI: https://doi.org/10.7554/eLife.38635.001

*For correspondence:
ehudc@ekmd.huji.ac.il

†These authors contributed equally to this work

Competing interests: The authors declare that no competing interests exist.

## Introduction

The view that aging is solely driven by stochastic events has changed as mounting evidences showed that certain, apparently independent, genetic and metabolic modulations, slow aging and extend lifespans of various organisms. Dietary restriction (DR), reduced activity of the insulin/IGF signaling pathway (IIS) or of the mitochondrial electron transport chain (ETC), and removal of germ cells (*Kenyon, 2005*), all slow the pace of aging. Among these, most prominent is IIS reduction, which extends lifespan and elevates stress resistance of worms (*Kenyon et al., 1993*), mice (*Holzenberger et al., 2003*), and presumably humans (*Suh et al., 2008*). In the nematode *Caenorhabditis elegans (C. elegans)*, the sole insulin/IGF receptor, DAF-2, initiates a signaling cascade that negatively regulates the activity of at least three transcription factors by modulating phosphorylation. Direct phosphorylation of DAF-16/FOXO (*Lee et al., 2001*) and of SKN-1/NRF (*Tullet et al., 2008*) prevents these factors from entering the nucleus and from regulating their target genes. Similarly, the IIS inhibits the phosphorylation of DDL-1, which retains the Heat Shock Factor 1 (HSF-1) in the cytosol (*Chiang et al., 2012*). Thus, IIS reduction by *daf-2* RNA interference (RNAi) or by mutation,

**eLife digest** Aging may seem inescapable, but there are many factors, from diet to genetic mutations, that can affect this process. In fact, scientists have started to uncover the mechanisms that control and influence this slow decline. For example, in the small worm *Caenorhabditis elegans*, removing the germs cells – which give rise to eggs – extends the lifespan. Similarly, interfering with the activity of the Insulin/IGF-1 signaling (IIS) pathway leads to a longer life for the animals. However, it is unclear whether these two mechanisms work together, or if they operate in parallel.

To explore this, Moll, Roitenberg et al. first looked at how the IIS pathway regulates a type of protein modification known as SUMOylation in *C. elegans*. Reducing the activity of the IIS pathway slowed down aging in the worms. It also decreased the levels of SUMOylation of certain proteins, including CAR-1, which is found in the structures that produce germ cells. Further experiments showed that stopping the SUMOylation of CAR-1 extended the lifespan of the animals. In fact, replacing the protein with a mutated version of CAR-1 that cannot accept the SUMO element makes the worms live longer and resist a toxic protein that causes Alzheimer's disease in humans. These results therefore show that, in *C. elegans*, the IIS pathway and a mechanism that involves CAR-1 in germ cells work together to determine the pace of aging. Further studies are now needed to dissect how the IIS pathway influences SUMOylation, and whether the findings hold true in mammals.
DOI: https://doi.org/10.7554/eLife.38635.002

hyperactivates its downstream transcription factors, creating long-lived worms (*Kenyon, 2005*). IIS reduction also elevates resistance to a variety of stresses including heat (*Lithgow et al., 1995*), ultraviolet (UV) radiation (*Murakami and Johnson, 1996*), and pathogenic bacteria (*Singh and Aballay, 2006*). In addition, IIS reduction protects worms and mice from toxic aggregation (proteotoxicity) of various neurodegeneration-causing proteins (reviewed in *Carvalhal Marques et al., 2015*). Finally, IIS reduction also modulates reproduction and egg-laying patterns, as knocking down *daf-2* by RNAi, reduces the worm's brood size but extends the reproduction period (*Dillin et al., 2002*). Although it was shown that the IIS is locally neutralized in germ cells (*Narbonne et al., 2015*) and that DAF-2 responds to food availability by modulating oogenesis through RAS-ERK signaling (*Lopez et al., 2013*), whether changes in post-translational modifications are involved in the IIS-mediated control of reproduction, is only partially understood. Moreover, despite the evidences that the ablation of germ cells extends lifespan (*Hsin and Kenyon, 1999*) and promotes proteostasis in *C. elegans* (*Shemesh et al., 2013*), it is unclear how the aging-regulating mechanisms downstream of the IIS and those that are activated by the reproduction system are linked, and whether post-translational modifications play roles in the orchestration of these mechanisms.

SUMOylation is a post-translational modification involving a reversible covalent attachment of a *s*mall *u*biquitin-like *mo*difier (SUMO) to specific lysine residues of proteins (*Melchior, 2000*). While mammals ubiquitously express three forms of SUMO (SUMO-1, 2, and 3), *C. elegans* expresses only one SUMO-encoding gene, *smo-1* that encodes a polypeptide of 91 amino acids with a predicted molecular weight of 10.2 kDa (*Choudhury and Li, 1997*). SUMOylation controls various biological processes and plays important roles in development and survival (*Johnson, 2004*). Among other functions, *smo-1* is critically needed for germline development and fertility of the nematode (*Broday, 2017*).

Here, we examined whether IIS activity controls SUMOylation of *C. elegans*' proteins and if this post-translational modification plays roles in aging-associated functions of this pathway. To address this, we compared global SUMOylation patterns of proteins that were extracted from untreated and from *daf-2* RNAi-treated animals, and found that among other modulations, IIS reduction lowers the SUMOylation rate of the protein CAR-1 (Cytokinesis/Apoptosis/RNA-binding protein 1) but has no effect on the expression level of *car-1*. CAR-1 is an RNA-binding protein, which acts in association with the RNA helicase, CGH-1 in the germline (*Audhya et al., 2005*). The knockdown of *car-1* increases the levels of GLP-1 during late oogenesis (*Noble et al., 2008*), thereby leading to germ cell death and to defective embryonic cytokinesis (*Boag et al., 2005*; *Squirrell et al., 2006*). We show that knocking down *car-1* shortens lifespan and enhances proteotoxicity in model worms. On the contrary, the expression of a mutant CAR-1, which cannot be SUMOylated on lysine residue 185

(K185), extends lifespan and promotes proteostasis. These effects are conferred, at least partially, through the GLP-1 axis in a DAF-16-dependent manner, but probably also through an additional, DAF-16-independent pathway. Interestingly, we found that GLP-1 positively controls the expression of *car-1* establishing a regulatory circuit. Our findings unveil a novel link between the reproductive system and the IIS, demonstrating that one downstream arm of this pathway regulates certain aspects of aging through the SUMOylation of CAR-1.

## Results

### IIS reduction results in differential protein SUMOylation in *C. elegans*

In order to test whether IIS reduction affects global protein SUMOylation in *C. elegans,* we employed three worm strains: wild-type animals (strain N2) and two conditionally sterile nematode strains: CF512 and CF1903, all exhibit natural IIS activity. CF512 animals become sterile when exposed to 25°C during development, as they cannot produce sperm. CF1903 animals harbor a temperature-sensitive *glp-1* mutant that renders them sterile upon exposure to 25°C during development (*Arantes-Oliveira et al., 2002*). Using these conditionally sterile worm strains, we could compare protein SUMOylation in adult tissues with no background from developing embryos. Eggs of all worm strains were extracted from animals that were grown in 15°C, and placed on plates that were seeded with either control bacteria, harboring the empty RNAi vector (EV), or with *daf-2* RNAi expressing bacteria. The plates were incubated at 25°C for 48 hr, transferred to 20°C for additional 24 hr and the worms were harvested at day 1 of adulthood. As expected, wild-type worms were fertile while CF512 and CF1903 worms were sterile (*Figure 1—figure supplement 1*). Global protein SUMOylation patterns were determined by western blot (WB) analysis, using an anti SUMO antibody. Our results indicated that IIS reduction modulates the patterns of SUMOylation in all three worm strains (*Figure 1A*), as the SUMOylation levels of several proteins were increased (arrowheads) and of others were decreased (arrows) upon treatment with *daf-2* RNAi. Differences in SUMOylation patterns between strains suggest strain-specific protein SUMOylation.

To identify proteins which are differentially SUMOylated upon IIS reduction, we used worms that were deprived of the endogenous *smo-1* gene and express a dually tagged *smo-1* transgene instead (*His-Flag-smo-1*, strain NX25 [*Pferdehirt and Meyer, 2013*]). NX25 worms were treated from hatching to day 1 of adulthood with *daf-2* RNAi or left untreated (EV), harvested and SUMOylated proteins were pulled-down by tandem-purification procedure (*Figure 1B* and *Figure 1—figure supplement 2*). Sediment proteins were analyzed by quantitative Mass Spectrometry (*Supplementary file 1*, full data set can be accessed at http://www.ebi.ac.uk/pride project ID: PXD010011). Our analysis showed that, among other affected proteins, the SUMOylation of CAR-1 is approximately threefold lower in *daf-2* RNAi-treated worms compared to the levels observed in untreated NX25 animals.

### IIS reduction lessens the SUMOylation of CAR-1

To further examine whether the IIS governs the rate of CAR-1 SUMOylation, we utilized worms that express CAR-1 fused to the green fluorescent protein (GFP) under the regulation of the *pie-1* promoter (strain WH346, GFP-CAR-1 [*Squirrell et al., 2006*]). These nematodes were used due to the high efficiency and specificity of GFP pulldown. The animals were developed on EV or *daf-2* RNAi bacteria, harvested at day 1 of adulthood and GFP-CAR-1 was immuno-precipitated using a GFP antibody, and blotted by an anti SUMO antibody. The intensities of two bands were remarkably higher in homogenates of untreated worms (EV) compared to homogenates of *daf-2* RNAi-treated worms (*Figure 1C*, arrows). One band migrated as a ∼ 60 kDa protein, the size corresponding to the mono SUMOylated GFP-CAR-1. The other band migrated as a protein of approximately 250 kDa, suggesting that SUMOylated GFP-CAR-1 is a component of a highly stable protein complex, perhaps with the RNA helicase CGH-1 (*Boag et al., 2005*). This complex appears to be less abundant, or less SUMOylated, in worms that exhibit low IIS activity. To compare the total amounts of GFP-CAR-1 in this pulldown experiment, we re-exposed the blot to an anti GFP antibody and found no difference in the quantities of GFP-CAR-1 molecules which migrated as a protein of approximately 50 kDa (*Figure 1D*).

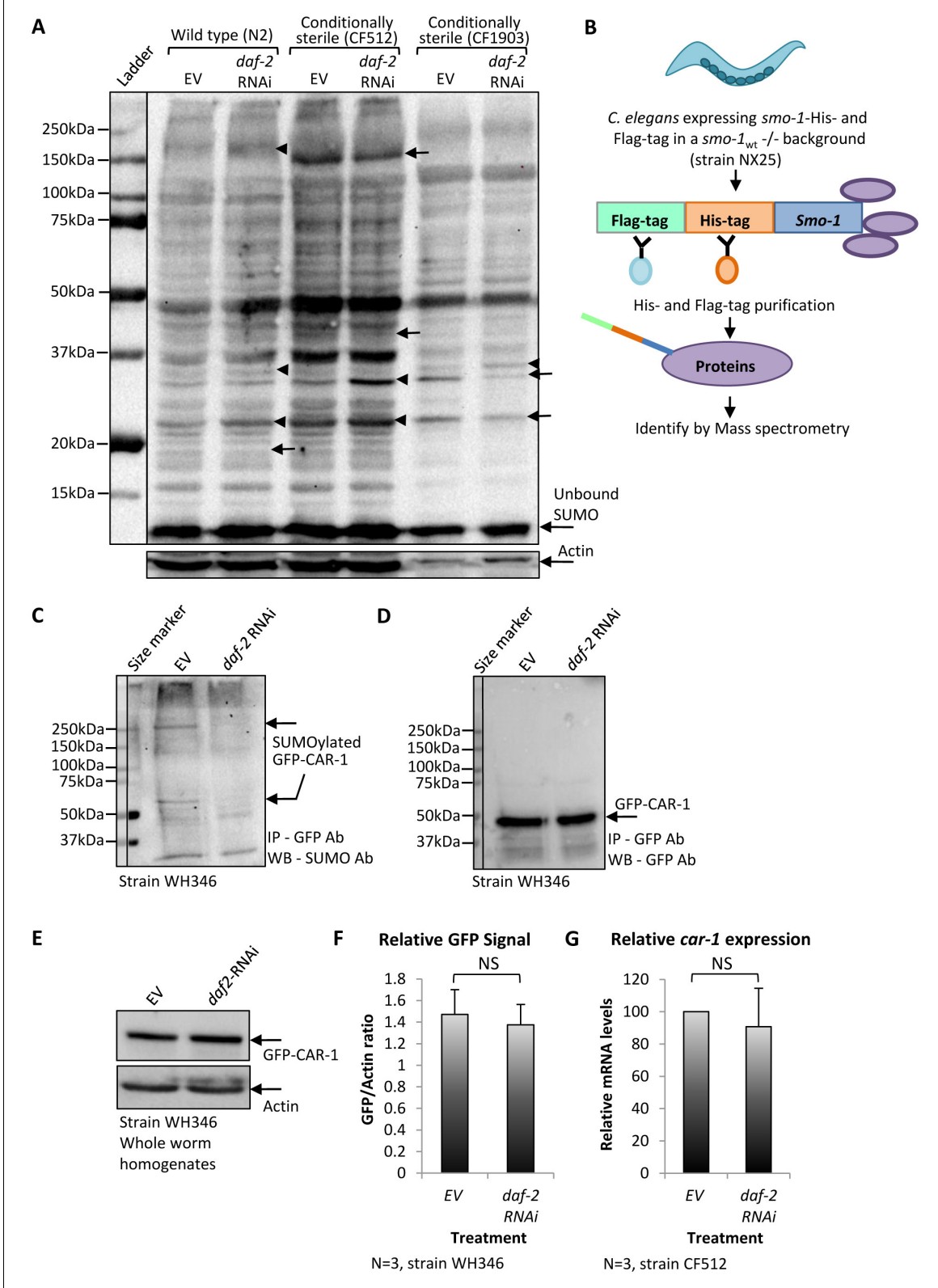

**Figure 1.** The knockdown of daf-2 modulates the SUMOylation of CAR-1 in *C.elegans*. (**A**) Global protein SUMOylation patterns in homogenates of *daf-2* RNAi-treated and untreated wild-type (N2), CF512 and CF1903 worms were compared by western blot using an anti-SUMO antibody. In all three worm strains, several proteins exhibit enhanced levels of SUMOylation upon the knockdown of *daf-2* (arrowheads) and others show decreased levels (arrows). (**B**) Schematic illustration of pulldown procedure to isolate covalently SUMOylated proteins from NX25 animals that express His-Flag-smo-1 in

*Figure 1 continued on next page*

*Figure 1 continued*

a *smo-1* knockout background. Covalently SUMOylated proteins were pulled down and identified by mass spectrometry. (C) Nematodes expressing GFP-tagged CAR-1 (strain WH346) were treated with *daf-2* RNAi or left untreated (EV), harvested at day 1 of adulthood and GFP-CAR-1 was immune-precipitated by a GFP antibody and blotted using a SUMO antibody. *daf-2* RNAi treatment reduced the level of SUMOylated GFP-CAR-1 that migrated as two bands. One was migrating as a protein of ~ 60 kDa and the other as a protein of ~ 250 kDa. (D) Reblotting the membrane with a GFP antibody showed that *daf-2* RNAi treatment had no effect on the amounts of the precipitated GFP-CAR-1 protein (this blot serves as a loading control for C). (E) WH346 worms were either grown on control bacteria (EV) or on *daf-2* RNAi bacteria, harvested at day 1 of adulthood and total GFP-CAR-1 amounts in the worm homogenates were analyzed by a western blot. No difference in the total levels of GFP-CAR-1 was observed. (F) Comparison of CAR-1-GFP signals in three independent experiments as in E. G. CF512 worms that were either treated with *daf-2* RNAi or left untreated (EV) express similar levels of *car-1* as measured by quantitative real-time PCR.

DOI: https://doi.org/10.7554/eLife.38635.003

The following figure supplements are available for figure 1:

**Figure supplement 1.** A Wild-type (N2) and two conditionally sterile worms: *rrf-3/fer-1* mut (CF512) and *glp-1* loss-of-function mutation (CF1903) that were used for the SUMOylation blot (*Figure 1A*).
DOI: https://doi.org/10.7554/eLife.38635.004
**Figure supplement 2.** (A–C) NX25 worms expressing *smo-1* tagged to a His- and Flag-tag in a *smo-1* null background, were grown on control (EV) or on *daf-2* RNAi bacteria.
DOI: https://doi.org/10.7554/eLife.38635.005

We next tested whether IIS reduction destabilizes the GFP-CAR-1 protein. WH346 worms were cultured from hatching on EV or *daf-2* RNAi bacteria, homogenized at day 1 of adulthood and WB analysis using an anti-GFP antibody was utilized to compare the relative levels of GFP-CAR-1. Our results showed similar amounts of GFP-CAR-1 in *daf-2* RNAi-treated and untreated worms (*Figure 1E*). Quantification of the GFP-CAR-1 signals in three independent repeats of this WB experiment confirmed that IIS reduction does not significantly changes the levels of this chimeric protein (*Figure 1F*). Finally, we tested the possibility that the lower level of SUMOylated CAR-1, observed in *daf-2* RNAi-treated worms (*Figure 1C*), stems from the regulation of *car-1* expression by the IIS. To address this, we employed CF512 worms, quantitative real-time PCR (qPCR) and *car-1*-specific primers and found no significant difference in the expression levels of *car-1* in *daf-2* RNAi-treated and untreated worms (*Figure 1G*).

Taken together, our observations indicate that the IIS modulates the SUMOylation of a sub-population of CAR-1 molecules and show that this signaling pathway affects neither the level of *car-1* expression nor the amounts of CAR-1 protein within the worm population.

## The roles of CAR-1 in the regulation of lifespan

Previous observations regarding the role of CAR-1 as a negative regulator of *glp-1* expression in germ cells (*Noble et al., 2008*), the well-documented effects of *glp-1* on aging (*Arantes-Oliveira et al., 2002*), and our findings of IIS-mediated SUMOylation of CAR-1 (*Figure 1*), have led us to speculate that CAR-1 is involved in the regulation of lifespan. To test this hypothesis, we compared the lifespans of wild-type worms and of nematodes that are *car-1* null. To obtain nematodes that lack *car-1*, we used worms that carry only one copy of the gene (strain WH377) and selected for progeny that lack both copies of *car-1* (*car-1* knockout worms are sterile). Lifespans of *car-1* knock-out worms were found to be significantly shorter than these of wild-type animals (strain N2) (*Figure 2A*, *Supplementary file 2*, mean lifespans (LS) of 14.81 ± 0.41 and 17.56 ± 0.52 days, respectively, p<0.001). A parallel experiment, using CF512 worms and RNAi towards *car-1* or *daf-16*, showed similar lifespan shortening by *car-1* RNAi (*Figure 2—figure supplement 1A*, and *Supplementary file 3*, mean LS of 14.87 ± 0.48 (*car-1* RNAi) and 18.01 ± 0.63 (EV) days, p<0.001). Nevertheless, the *car-1* RNAi-mediated lifespan shortening effect was less prominent than that of *daf-16* RNAi (*Figure 2—figure supplement 1A*, mean LS of 12.19 ± 0.44 days).

Next, we asked whether CAR-1 is needed for the full longevity phenotype of nematodes that carry a weak *daf-2* allele (*e1370*, strain CB1370). The worms were either grown from hatching on *daf-16*, *car-1* RNAi, or left untreated (EV) and their lifespans were recorded. The knockdown of *car-1* significantly shortened the lifespan of *daf-2* mutant worms, compared to untreated animals (*Figure 2B*, *Supplementary file 2*, mean LS of 40.02 ± 1.40 (*car-1* RNAi) and 50.52 ± 1.38 (EV) days, respectively, p<0.001). This effect was less prominent than that of *daf-16* knockdown (mean LS of

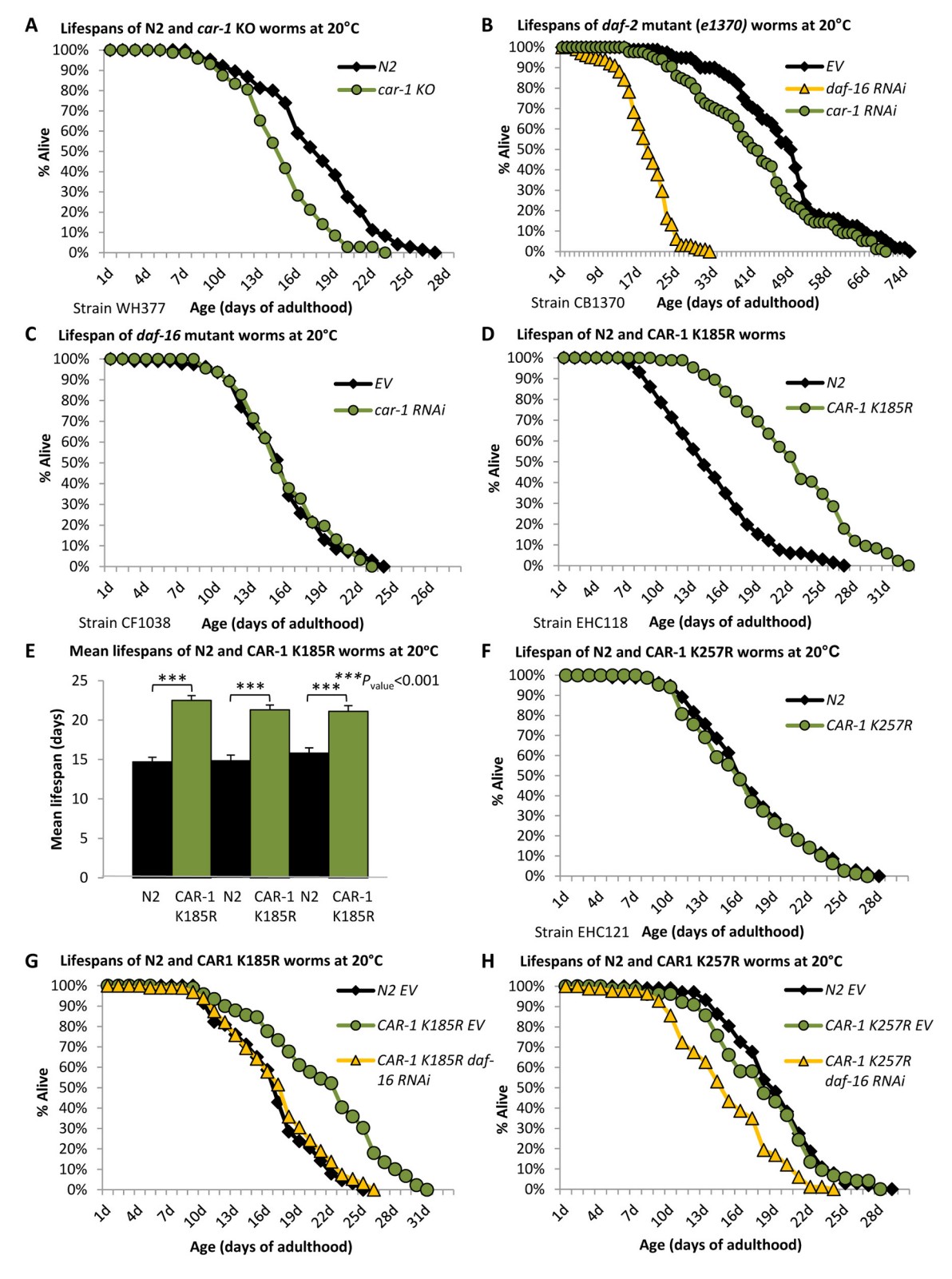

**Figure 2.** *car-1* regulates lifespan through the germline of *C.elegans*. (**A**) The knockout of *car-1* (strain WH377) shortens lifespan compared to wild-type worms (N2) (mean LS of 14.81 ± 0.41 and 17.56 ± 0.52 days, p<0.001). (**B**) *daf-2* mutant (*e1370*) worms were treated with *car-1*, *daf-16* RNAi or left untreated (EV) and lifespans were followed. *car-1* RNAi as well as *daf-16* RNAi, shortened lifespan compared to control worms, however, the lifespan-shortening effect of *daf-16* RNAi was more prominent than that of *car-1* RNAi (mean LS of 17.56 ± 0.56, 40.02 ± 1.40 and 50.52 ± 1.38 days, respectively
*Figure 2 continued on next page*

*Figure 2 continued*

(p<0.004, EV vs. *car-1* RNAi)). (C) The lifespans of *car-1* RNAi-treated and untreated *daf-16* mutant worms (strain CF1038) were indistinguishable (mean LS of 15.74 ± 0.47 and 15.37 ± 0.48 days, p=0.3). (D–E) Worms that express CAR-1 K185R (EHC118) live approximately 53% longer than their wild-type counterparts (mean LS of 22.51 ± 0.60 and 14.70 ± 0.59 days respectively, p<0.001), as shown by a representative experiment (D) and a summary of three independent experiments (E). (F) Worms that express CAR-1 K257R (EHC121) and wild-type animals have nearly identical lifespans (16.32 ± 0.54 and 16.84 ± 0.56 days, respectively, p=0.25). (G) The longevity of CAR-1 K185R expressing worms is DAF-16-dependent (mean LS of 18.32 ± 0.47 (*daf-16* RNAi) and 21.73 ± 0.61 (EV) days, p<0.001). However, *daf-16* RNAi-treated CAR-1 K185R and wild-type worms had nearly identical lifespans (16.90 ± 0.63 and 16.67 ± 0.52 respectively, p=0.42). (H) *daf-16* RNAi shortens the lifespans of CAR-1 K257R expressing worms (mean LS of 14.95 ± 0.47 days) compared to untreated wild-type animals (18.09 ± 0.41 days, p<0.001).
DOI: https://doi.org/10.7554/eLife.38635.006

The following figure supplements are available for figure 2:

**Figure supplement 1.** The knockdown of *car-1* shortens lifespan.
DOI: https://doi.org/10.7554/eLife.38635.007

**Figure supplement 2.** The computational GPS-SUMO tool identified the SUMOylation consensus sequences in CAR-1 pointing at two lysine residues, 185 and 257 as potential SUMOylation sites.
DOI: https://doi.org/10.7554/eLife.38635.008

**Figure supplement 3.** Worms that over-express either WT CAR-1 (EHC117), CAR-1 K185R (EHC118) or CAR-1 K257R (EHC121) were harvested at day 1 of adulthood, homogenized and equal protein amounts were subjected to IP using an HA antibody (all exogenous CAR-1 proteins are tagged with a double HA tag).
DOI: https://doi.org/10.7554/eLife.38635.009

**Figure supplement 4.** CAR-1 K185R extends lifespan.
DOI: https://doi.org/10.7554/eLife.38635.010

17.56 ± 0.56 days). A similar trend was seen when an additional *daf-2* mutant worm strain (*e1368*) was used (*Figure 2—figure supplement 1B* and *Supplementary file 2*). Yet, the lifespan reduction that we observed among untreated and *car-1* RNAi-treated *daf-2* (*e1370*) mutant worms, which was similar to the difference observed among wild-type and *car-1* knockout animals (*Figure 2A*), questioned the notion that CAR-1 is involved in IIS-mediated regulation of lifespan. Thus, these results suggest that the SUMOylation of CAR-1 by the IIS may be involved in other functions of this signaling pathway.

To further characterize the roles of *car-1* as a regulator of lifespan, we tested whether the knockdown of *car-1* affects the lifespan of *daf-16* mutant animals (strain CF1038), and found that *car-1* RNAi had no effect on the lifespans of these worms (*Figure 2C* and *Supplementary file 2*). This observation implies that the lifespan regulatory functions of *car-1* are DAF-16 dependent. However, since DAF-16 is involved in several longevity-controlling mechanisms (*Hsin and Kenyon, 1999*), we further tested whether CAR-1 is mechanistically linked to IIS.

Since IIS reduction lowers CAR-1 SUMOylation levels (*Figure 1C*), we sought to test whether the SUMOylation state of CAR-1 affects lifespan. To directly address this, we used the computational tool GPS-SUMO (*Zhao et al., 2014*), for identifying SUMOylation consensus motifs in the sequence of CAR-1. Two lysine residues, K185 and K257 were found to be located within predicted SUMOylation motifs. Among the two, K185 is more likely to serve as a SUMOylation site (*Figure 2—figure supplement 2*). If SUMOylation of K185 or K257 reduces CAR-1 activity and shortens lifespan, it was expected that the overexpression of a SUMOylation-resistant CAR-1 mutant would extend lifespan. To test this hypothesis, we created worms that over-express mutated *car-1* genes in which either K185 or K257 were substituted with arginine (CAR-1 K185R (EHC118) and CAR-1 K257R (EHC121), both strains also express the endogenous, wild-type *car-1*). These mutations prevent potential SUMOylation but maintain the hydrophobicity of the protein. To control for the effect of *car-1* overexpression on lifespan, we also created worms that over-express the wild-type *car-1* gene (strain EHC117). The three exogenous *car-1* genes were fused to an N-terminal double HA tag, and their expression levels were controlled by the *car-1* promoter. To compare the levels of SUMOylated CAR-1 in these worm strains, we homogenized young adult worms of the three strains and subjected equal amounts of protein to IP. Using an HA antibody, we pulled down CAR-1, separated total proteins of each worm strain and blotted SUMOylated CAR-1 by a SUMO antibody. Our results (*Figure 2—figure supplement 3*) show that worms that express the WT CAR-1 contain much higher levels of SUMOylated CAR-1 compared to their counterparts that overexpress either CAR-1 K185R

or CAR-1 K257R, indicating that these are SUMOylation sites. Our results also show that CAR-1 is SUMOylated on more than one site, as the substitution of either K185 or of K257 with arginine, did not abolish SUMOylation of the protein.

Performing lifespan assays we found that the over-expression of CAR-1 K185R significantly extended the worms' lifespans compared to those of wild-type animals (*Figure 2D*, *Supplementary file 2*, mean LS of 22.51 ± 0.60 and 14.70 ± 0.59, p<0.001). Three independent repeats confirmed the significance of this phenotype (*Figure 2E*, *Figure 2—figure supplement 4, A and B*, and *Supplementary file 2*). In contrast, no lifespan extension was observed in worms that overexpress CAR-1 K257R (*Figure 2F* and *Supplementary file 3*) or wild-type CAR-1 (*Figure 2—figure supplement 4C* and *Supplementary file 3*), indicating that the SUMOylation of CAR-1 on lysine 185, but not on lysine 257, plays a role in lifespan determination.

To examine whether the lifespan-extending mechanism that is activated by CAR-1 K185R is DAF-16-dependent, we utilized CAR-1 K185R worms (of a second clone). The worms were either treated with *daf-16* RNAi or left untreated (EV), and their lifespans were monitored. Surprisingly, our results (*Figure 2G*) show that *daf-16* RNAi-treated CAR-1 K185R worms and wild-type (N2) animals, exhibited indistinguishable lifespans (see also *Supplementary file 2,3*). In contrast, *daf-16* RNAi-treated CAR-1 K257R worms (*Figure 2H*) and animals that over-express the wild-type CAR-1 and fed with *daf-16* RNAi bacteria (*Figure 2—figure supplement 4C* and *Supplementary file 3*) had shorter lifespans compared to their wild-type (N2) counterparts. These results suggest that CAR-1 also regulates lifespan by a DAF-16-independent mechanism.

Taken together, our observations indicate that CAR-1 is needed for wild-type worms to live their natural lifespan and for *daf-2* mutant animals to exhibit their full longevity phenotype. They also indicate that SUMOylation of K185 plays a role in the regulation of lifespan. Interestingly, DAF-16 is needed for CAR-1 K185R to extend lifespan; however, the knockdown of *daf-16* reduces the lifespans of CAR-1 K185R-expressing animals to be similar to these of wild-type worms, but not shorter as expected.

## The mechanisms of CAR-1-mediated lifespan regulation

One possible explanation to the lifespan shortening effect of *car-1* RNAi, and the longevity conferred by CAR-1 K185R, suggests that CAR-1 modulates lifespan by negatively regulating the activity of GLP-1. Accordingly, knocking down *car-1* by RNAi is expected to hyperactivate GLP-1 and shorten lifespan, whereas the expression of the SUMOylation-resistant, hyperactive CAR-1 K185R, is expected to lower the activity of GLP-1, thereby extending lifespan. To scrutinize this hypothesis, we utilized CF1903 worms. If the lifespan shortening effect of *car-1* RNAi is mediated by hyper-activating GLP-1, it was expected that the knockdown of *car-1* would not shorten the long lifespans of these worms, which lack functional GLP-1. CF1903 worms were either grown throughout life on control bacteria (EV), or treated with RNAi towards *daf-16* or *car-1*, and lifespans were recorded (in this experiment the worms were developed at 25°C and transferred to 20°C at day 1 of adulthood). While *daf-16* RNAi-treated animals had shorter lifespans (*Figure 3A*, *Supplementary file 2*, mean LS of 10.71 ± 0.33 days, p<0.001), untreated and *car-1* RNAi-treated worms had very similar lifespans (mean LS of 18.13 ± 0.85 (EV) and 17.76 ± 0.72 (*car-1* RNAi) days, respectively, p=0.37).

These results show that *car-1* RNAi does not affect lifespan in the absence of functional GLP-1, and support the theme that CAR-1 modulates lifespan by controlling the activity of GLP-1.

To further assess the hypothesis that CAR-1 affects lifespan through the GLP-1 axis, we utilized worms that carry a mutant *kri-1* gene (strain CF2052 (ok1251)). *kri-1* is essential for the mediation of longevity by germ cell ablation but not by IIS reduction (*Berman and Kenyon, 2006*). Thus, if *car-1* affects lifespan through the modulation of GLP-1 activity, *car-1* RNAi is expected not to affect the lifespan of *kri-1* mutant worms. CF2052 nematodes were treated with *daf-16* or *car-1* RNAi and lifespans were recorded. Although *daf-16* RNAi-treated CF2052 animals exhibited a short mean lifespan (13.37 ± 0.26 days, p<0.001), untreated and *car-1* RNAi-treated worms had similar mean lifespans of 16.59 ± 0.43 and 15.96 ± 0.39 days, respectively (*Figure 3B*, p=0.14, and *Supplementary file 2*). Together these observations support the theme that *car-1* governs lifespan, at least partially, through a *glp-1*-controlled mechanism.

The possible involvement of *car-1* in the regulation of lifespan through an additional, *daf-16*-independent mechanism has led us to ask whether the knockdown of *car-1* is linked to the aging-regulating pathway downstream of the transforming growth factor β (TGF-β). This pathway converges with

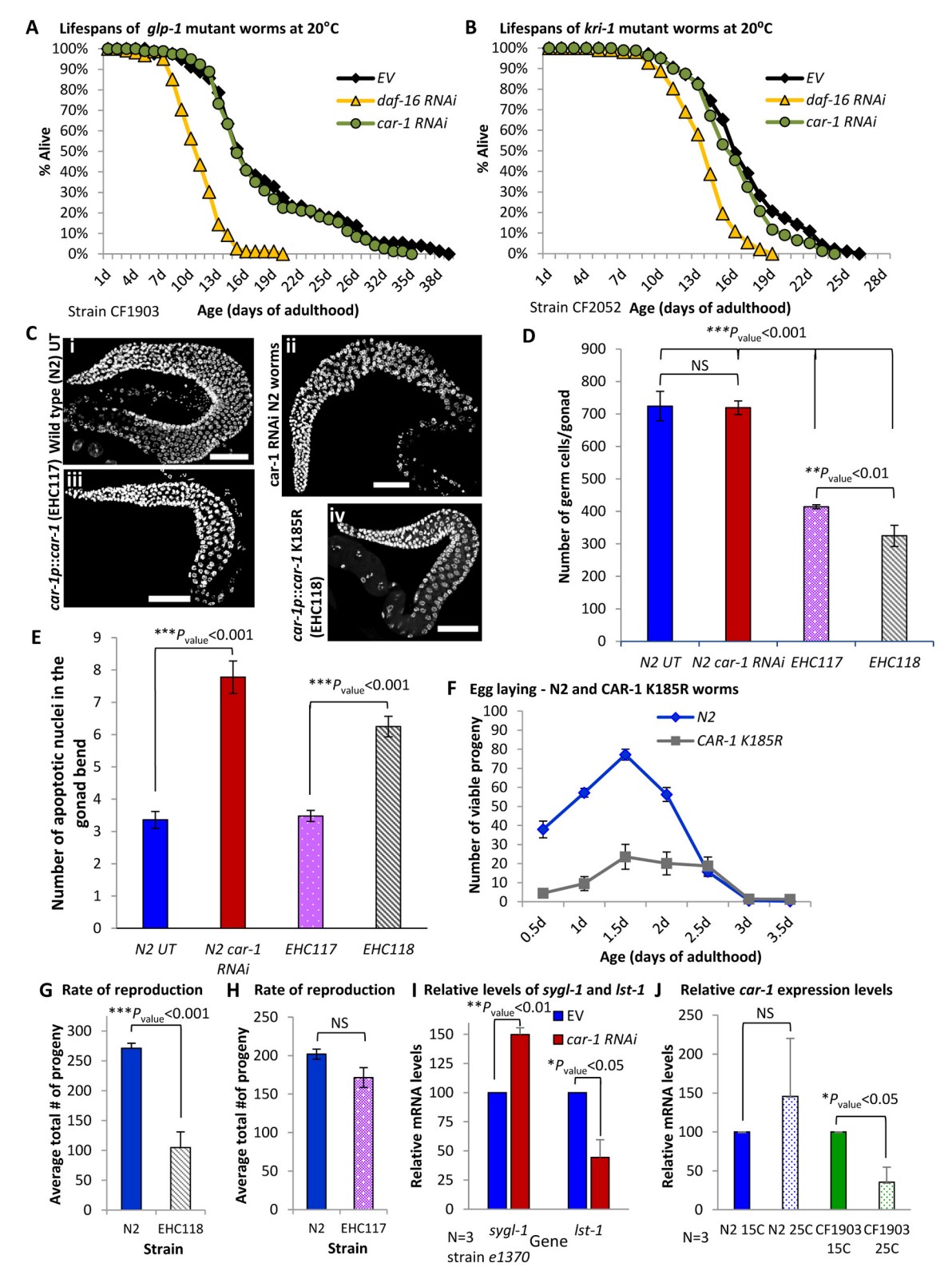

**Figure 3.** The knockdown of *car-1* modulates the activity of *glp-1*. (**A**) Untreated (EV) and *car-1* RNAi-treated, long-lived *glp-1* mutant worms (strain CF1903) show no difference in lifespans (mean LS of 17.76 ± 0.72 and 18.13 ± 0.85 days, respectively p=0.37). In contrast, *daf-16* RNAi reduced the lifespan of these animals (10.71 ± 0.0.33 days, p<0.001). (**B**) *car-1* RNAi treatment has no significant effect on the lifespans of *kri-1* mutant worms (mean LS of 16.59 ± 0.43 (EV) and 15.96 ± 0.39 (*car-1* RNAi), p=0.14). In contrast, *daf-16* RNAi shortened the lifespans of these animals (mean LS of 13.37 ± 0.26

*Figure 3 continued on next page*

*Figure 3 continued*

days (*daf-16* RNAi), p<0.001. (**C**) DAPI stained image of gonads of worms of the indicated genotypes. Bar = 50 mm. (**D**) The over expression of CAR-1 (strain EHC117) or of the mutated K185R CAR-1 (EHC118) resulted in significantly reduced number of germ cells in the worms' gonads. The expression of K185R CAR-1 (in EHC118 animals) reduces the number of germ cells by ~ 22% compared to the number that was observed in worms that express the wild-type CAR-1 (EHC117) (p<0.01). (**E**) Quantification of germline apoptosis by acridine orange staining. *car-1* RNAi elevates the average number of apoptotic cells in the gonads of wild-type worms by ~ 2.5-fold. The average numbers of apoptotic cells in the gonads of untreated N2 and of EHC117 worms are nearly identical (3.36 and 3.48, respectively). In contrast, the expression of K185R CAR-1 elevates the average number of apoptotic cells by ~ 80% compared to untreated N2 and EHC117 (p<0.0001, bars represent ± SEM). (**F–H**) Worms expressing CAR-1 K185R (EHC118) have reduced number of progeny compared to N2 animals (**F**). The average total number of progeny of EHC118 animals was 105 while control animals had an average of 271 offspring (**G**). No significant difference in the brood size of wild-type (N2) and EHC117 worms (**H**). (**I**) The knockdown of *car-1* increases the expression levels of *sygl-1* compared to the levels detected in untreated *daf-2* (*e1370*) but decreases the levels of *lst-1*. (**J**) The knockdown of *glp-1* (in CF1903 worms that were exposed to 25°C) lowers the expression levels of *car-1*. No such effect was observed in wild-type animals.

DOI: https://doi.org/10.7554/eLife.38635.011

The following figure supplements are available for figure 3:

**Figure supplement 1.** CAR-1 does not appear to function through signals from the somatic gonad.
DOI: https://doi.org/10.7554/eLife.38635.012
**Figure supplement 2.** The knockdown of *car-1* affects egg-laying patterns of *daf-2*, *daf-16* and *kri-1* mutants but not of *daf-9* and *daf-12* mutant animals.
DOI: https://doi.org/10.7554/eLife.38635.013
**Figure supplement 3.** (A–B) Wild-type (N2) and CF1903 worms were grown in either 15 or 25°C (CF1903 worms lack *glp-1* when grown in 25°C).
DOI: https://doi.org/10.7554/eLife.38635.014
**Figure supplement 4.** The knockdown of *car-1* by RNAi tends to elevate the expression level of *sygl-1* in N2 worms.
DOI: https://doi.org/10.7554/eLife.38635.015

the IIS at the nuclear hormone receptor DAF-12, whose activation is regulated by the cytochrome P450 enzyme, DAF-9 (*Gerisch et al., 2001*). To test this, we followed the lifespans of *daf-9* (strain CF2531) and of *daf-12* (strain AA86) mutant worms that were either treated with *daf-16*, *car-1* RNAi or left untreated, and found that the lifespans of both worm strains were shortened by *daf-16* RNAi as well as by the knockdown of *car-1* (*Figure 3—figure supplement 1, A and B*, *Supplementary file 4*). The similar rates of lifespan shortening that resulted from the knockdown of *car-1* in wild-type worms (*Figure 2A*), *daf-9* and *daf-12* mutant worms, strongly suggest that *car-1* RNAi shortens lifespan by a *daf-9* and *daf-12*-independent mechanism.

## The roles of CAR-1 in GLP-1-mediated functions

Beside its roles in lifespan determination, GLP-1 is also involved in germ cells proliferation and reproduction (*Austin and Kimble, 1987*). Thus, we asked whether the modulation of *car-1* expression and activity modifies the amount of germ cells. To test this, we compared the numbers of germ cells in gonads of four groups of worms: (i) untreated, wild-type worms (strain N2), (ii) *car-1* RNAi-treated wild-type animals, (iii) worms that over-express the natural *car-1* (EHC117), and (iv) nematodes that over-express the CAR-1 K185R mutant (EHC118). Nuclei were stained with DAPI and germ cells were counted. We found that the knockdown of *car-1* by RNAi had no significant effect on the number of germ cells as untreated and *car-1* RNAi-treated animals had similar numbers of germ cells (*Figure 3*, C and D; average of 724 ± 45.25 and 719 ± 32.36 cells, respectively). In contrast, worms that over-express the natural CAR-1 (EHC117) had significantly less germ cells compared to untreated or *car-1* RNAi-treated worms (414.2 ± 6.3 cells, p<0.001). The over-expression of CAR-1 K185R resulted in further reduction in the number of germ cells (average of 324.8 ± 21.37 germ cells/gonad, p<0.01 compared to EHC117).

The observation that the knockdown of *car-1* shows no effect on the number of germ cells may emanate from an efficient SUMOylation-mediated inactivation of CAR-1 in wild-type worms. Thus, CAR-1 is less active and its knockdown has a small effect on the number of germ cells. In contrast, the over-expression of CAR-1 (in EHC117 worms), may exceeds the capacity of the SUMOylation mechanism and thus, hyper-activates CAR-1, which in turn lowers the activity of GLP-1, thereby reducing the number of germ cells. According to this explanation, the over-expression of the hyper-active CAR-1 K185R (EHC118) further suppresses the activity of GLP-1, resulting in an even lower number of germ cells.

The knockdown of *car-1* by RNAi was reported to enhance physiological apoptosis in hermaphrodites (*Boag et al., 2005*). Thus, we examined how knocking down *car-1*, or over expressing wild-type or the mutant CAR-1 K185R, affect the rate of apoptosis in the gonads. Our results (*Figure 3E*) indicate that, as shown previously (*Boag et al., 2005*), the knockdown of *car-1* elevates the number of apoptotic nuclei in the gonad by approximately 2.5 fold. A similar increase in the rate of apoptosis was seen in EHC118 worms, but not in EHC117 animals. These observations raise the question of how the effects of CAR-1 and its SUMOylation on the number of germ cells and of apoptotic nuclei, affect reproduction.

To address this we tested how the expression of wild-type or of CAR-1 K185R affects brood size by comparing the egg laying capabilities of N2, EHC117 and EHC118 worms. The animals were grown from hatching on control bacteria. At L4 larval stage, 12 animals of each strain were transferred onto new plates, one animal per plate. The worms were transferred onto new plates in 12 hr intervals and viable progeny were counted 48 hr thereafter. We found that EHC118 animals lay fewer eggs than control worms (*Figure 3*, F and G). While in total, N2 worms had an average of 271.1 progeny, each CAR-1 K185R worm had an average of 104.8 living offspring (*Figure 3G*). A small but not significant difference was observed among EHC117 and N2 worms, as on average EHC117 worms had 171.4 viable offspring and N2 worms had 201.9 (*Figure 3H*).

Our observations indicate that SUMOylation of CAR-1 on residue 185 is involved in the modulation of reproduction and suggest that this phenotype may be associated with the effects of IIS reduction on egg laying (*Dillin et al., 2002*). Thus, we examined how the knockdown of *car-1* affects the egg-laying pattern of worms that exhibit impaired IIS. To test this, we employed *daf-2* (*e1370*) mutant and *daf-16* mutant (*mu86*, strain CF1038) worms. *e1370* worms were treated with either *car-1* RNAi, *daf-16* RNAi, or were left untreated (EV), while CF1038 animals were treated with *car-1* RNAi or fed on control bacteria (EV). Egg-laying patterns were followed as described above. Our results (*Figure 3—figure supplement 2, A and B*) confirmed that untreated *e1370* worms had a much longer reproductive period compared to wild-type animals. However, the total number of progeny per untreated *daf-2* mutant worm was on average 95.7, much lower than that of wild-type nematodes (*Figure 3G*). Both phenotypes were largely rescued by the knockdown of *daf-16,* which shifted egg laying to early adulthood (*Figure 3—figure supplement 2A*) and restored the total number of eggs to the average of 255.4 (*Figure 3—figure supplement 2B*).

Surprisingly, RNAi-mediated knockdown of *car-1* resulted in nearly complete sterility of both *daf-2* and *daf-16* mutant worms (*Figure 3—figure supplement 2, A and B*). These observations show that worms that have impaired IIS activity are much more sensitive to the knockdown of *car-1* than wild-type animals, implying that this gene is involved in the control of reproduction by the IIS. Yet, this reduction in brood size may be partially due to additive effects of IIS reduction and knocking down *car-1*.

We also tested whether knocking down *car-1* influences the egg-laying patterns of *kri-1* mutant (CF2052) nematodes. As shown previously (*Dillin et al., 2002*), untreated wild-type worms laid the highest number of eggs between day 1 and 1.5 of adulthood (*Figure 3F*). In contrast, N2 worms that were treated with *car-1* RNAi laid the highest number of eggs at the beginning of their reproductive stage (day 0 to 0.5), and the number of eggs declined thereafter (*Figure 3—figure supplement 2, C and D*). While untreated N2 worms laid on average 244 eggs in total, their *car-1* RNAi-treated counterparts laid only 133 eggs (a reduction of ~ 45% (*Figure 3—figure supplement 2D*)). The reduced reproduction was surprising, as the knockdown of *car-1* by RNAi had no effect on the number of germ cells (*Figure 3*, C and D). Nevertheless, this reduction, which is consistent with a previous report (*Boag et al., 2005*), may be explained by the increase in the number of apoptotic cells in the gonads of these animals (*Figure 3E*). *kri-1* mutant worms (CF2052) laid fewer eggs than N2 animals, but *car-1* RNAi treatment further reduced the number of progeny of both strains. While untreated *kri-1* mutant worms laid an average of 116 eggs, *car-1* RNAi-treated worms of the same strain laid merely 10 eggs (*Figure 3—figure supplement 2, C and D*).

Unexpectedly, the knockdown of *car-1* affects neither egg-laying patterns, nor brood size of *daf-9* and *daf-12* mutant worms (*Figure 3—figure supplement 2, E and F*). These observations show that the egg-laying modulation that resulted from the knockdown of *car-1* by RNAi is dependent on the presence of functional *daf-9* and *daf-12,* linking CAR-1 also with the DAF-9/DAF-12 pathway.

Altogether, despite the similarity in the number of germ cells in *car-1* RNAi-treated and untreated N2 worms, the reduction in brood size of wild-type worms by *car-1* RNAi, strongly suggests that

CAR-1 regulates reproduction by GLP-1-dependent and GLP-1-independent mechanisms. In addition, our observation that the knockdown of *car-1* leads to nearly complete sterility of worms carrying weak *kri-1* or *daf-2* alleles or nonfunctional *daf-16,* suggests that CAR-1 is also a component of the reproduction-regulating mechanism downstream of the IIS.

## The levels of car-1 modulate the transcriptional activity of the GLP-1 pathway

To directly test whether *car-1* affects the transcriptional activity of the GLP-1 pathway, we asked how the knockdown of *car-1* affects the expression levels of the *glp-1*-target genes *sygl-1* and *lst-1* (*Shin et al., 2017*). First, we used N2 and CF1903 to confirm the regulatory roles of GLP-1 on the transcription of these genes. Worms of both strains were either grown at 15 or 25°C (to inactivate GLP-1 in CF1903 animals) and the expression levels of *sygl-1* and *lst-1* were determined by qPCR. While an increase in the expression levels of both genes was observed in N2 worms upon exposure to 25°C (a significant increase for *sygl*-1 and a non-significant trend for *lst*-1), the inactivation of GLP-1 in CF1903 worms, by exposing them to 25°C during development, resulted in a significant reduction in the expression of both, *sygl*-1 and *lst*-1 (*Figure 3—figure supplement 3, A and B*). These results indicate that GLP-1 positively regulates the expression of these two genes.

We next utilized *daf-2* (*e1370*) mutant animals to test how the knockdown of *car-1* affects the expression of *sygl-1* and *lst-1* in these nematodes. The worms were grown from hatching on EV or on *car-1* RNAi bacteria, harvested at day 1 of adulthood and gene expression levels were compared by qPCR. If CAR-1 is a negative regulator of GLP-1 that is negatively controlled by IIS-mediated SUMOylation, it is expected that CAR-1 is hyperactive in *daf-2* mutant worms. Accordingly, the knockdown of *car-1* by RNAi is predicted to activate GLP-1 and elevate the expression of *sygl-1* and *lst-1*. Indeed, we observed significantly elevated levels of *sygl-1* in *e1370* animals (*Figure 3I*). A non-significant elevation in the expression of *sygl-1* was also seen in *car-1* RNAi-treated N2 worms (*Figure 3—figure supplement 4*). This lack of significance may be explained by the low activity of CAR-1 in these worms, due to its SUMOylation. These results indicate that CAR-1 negatively controls the activity of GLP-1 as a transcriptional regulator of *sygl-1.*

Unexpectedly, the knockdown of *car-1* reduced the expression level of *lst-1* in *e1370* worms (*Figure 3I*) showing that CAR-1 could play opposing roles on the expression levels of GLP-1 target genes. This observation is consistent with the finding that in some cases, transcriptional co-factors affect the expression of some target genes but not of others (*Volovik et al., 2014b*), and show the complex regulatory relations between *car-1* and *glp-1.*

Since the IIS controls the expression levels of some of its components (*Alic et al., 2011*), we asked whether GLP-1 controls the expression of *car-1.* Using qPCR, CF1903, and wild-type worms, we found that CF1903 animals that were developed at 25°C and thus, lack functional GLP-1, have reduced *car-1* levels compared to their counterparts that were grown at 15°C. No significant difference in the expression of *car-1* was observed in wild-type worms (*Figure 3J*). This reduction of approximately 65% in the levels of *car-1*, shows that GLP-1 positively regulates the expression of *car-1*, and raises the question of whether CAR-1 also plays roles in another feature of the GLP-1-controlled mechanism, the maintenance of proteostasis.

## CAR-1 is involved in maintaining proteostasis

The known regulatory roles of *glp-1* on proteostasis (*Shemesh et al., 2013*) has led us to examine whether CAR-1 also controls proteotoxicity. To address this, we utilized worms that express the Alzheimer's disease associated, human $A\beta_{3-42}$ peptide (*McColl et al., 2009*), in their body wall muscles (strain CL2006, Aβ worms) (*Link, 1995*). The expression of Aβ causes progressive paralysis within the worm population, a phenotype that can be alleviated by the knockdown of *daf-2* (*Cohen et al., 2006*). Eggs of Aβ worms were placed on plates seeded with *daf-2* or *car-1* RNAi bacteria, or left untreated (EV). Rates of paralysis were followed up until day 12 of adulthood. While the knockdown of *daf-2* protected the worms from Aβ-mediated toxicity, animals that were treated with *car-1* RNAi exhibited higher rate of paralysis than untreated worms (*Figure 4A*). Five independent repeats confirmed the significance of this phenotype (*Figure 4B*).

We next examined whether CAR-1 is needed for *daf-2* RNAi-conferred protection from proteotoxicity, by performing paralysis assays using Aβ worms that were grown on different mixtures of

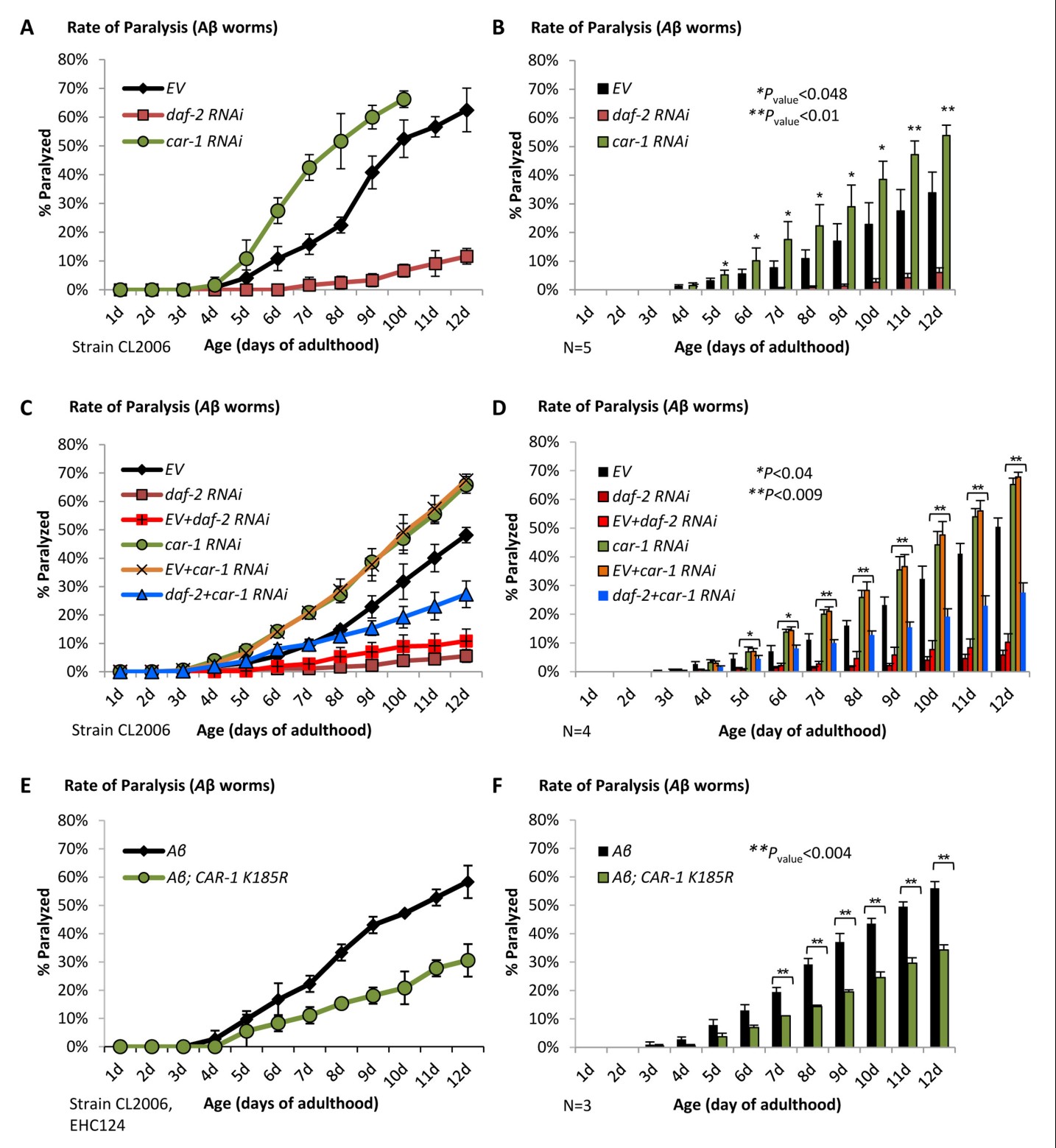

**Figure 4.** CAR-1 modulates proteostasis in *C.elegans*. (A–B) Worms expressing Aβ$_{3-42}$ in their body wall muscles were treated with *car-1*, *daf-2* RNAi or left untreated (EV). While *daf-2* RNAi protected from paralysis, *car-1* RNAi significantly increased paralysis (N = 5). (C–D) The dilutions of *car-1* RNAi (orange) or of *daf-2* RNAi (red) bacteria with control bacteria do not significantly change the effects of these treatments on Aβ-mediated paralysis. Concurrent knockdown of *daf-2* and *car-1* by RNAi only partially protects Aβ worms from paralysis (C, blue). The increased rate of paralysis after *car-1* RNAi treatment and the reduced paralysis after *car-1* and *daf-2* knockdown were significant compared to the level seen in untreated worms (EV) (N = 4,

*Figure 4 continued on next page*

*Figure 4 continued*

p<0.04) (**D**). (**E–F**) The expression of CAR-1 K185R in Aβ worms (strain EHC124) protects the animals from paralysis (**E**). Three independent experiments confirmed the significance of this observation (**F**) (p<0.004).

DOI: https://doi.org/10.7554/eLife.38635.016

The following figure supplement is available for figure 4:

**Figure supplement 1.** CAR-1 K185R but not CAR-1 K257R modulates proteotoxicity.

DOI: https://doi.org/10.7554/eLife.38635.017

RNAi bacterial strains. First, we checked if the dilution of bacteria expressing either *car-1* or *daf-2* RNAi with control bacteria (EV) reduces the effects of these treatments on paralysis. Worms that were solely treated with *car-1* RNAi and their counterparts that were fed with a mixture of *car-1* RNAi and EV bacteria exhibited very similar rates of paralysis over time. Likewise, similar protection from paralysis was seen in worms that were exclusively fed with *daf-2* RNAi bacteria and those which were fed with a mixture of *daf-2* RNAi and EV bacteria (*Figure 4*, C and D). These results show that the dilution of *car-1* and of *daf-2* RNAi bacteria with another bacterial strain does not significantly changes the effects of these treatments on proteostasis.

We next examined whether the mixture of *daf-2* and *car-1* RNAi treatments prevents IIS reduction from promoting its full counter-proteotoxic effect, and found that concurrent knockdown of these genes resulted in an enhanced rate of paralysis compared to the rate observed in animals that were treated solely with *daf-2* RNAi (*Figure 4*, C and D, blue, p<0.04). These findings imply that CAR-1 is needed for proteostasis-maintenance, downstream of the IIS however, the reduced paralysis rates that we observed in worms that were concomitantly treated with *daf-2* and *car-1* RNAi imply that the knockdown of *daf-2* protects from proteotoxicity by additional, CAR-1-independent mechanisms. Yet, it is also possible that knocking down *car-1*, may inflict damage by an IIS-independent mechanism. According to this notion, the observed cumulative paralytic effect results from both, *daf-2* RNAi-mediated protection and *car-1* RNAi-promoted damage. To directly address this hypothesis, we expressed CAR-1 K185R in Aβ worms (strain EHC124). If SUMOylation on K185 reduces the activity of CAR-1, it was expected that the expression of the SUMOylation-resistant CAR-1 K185R mutant would protect the animals from Aβ toxicity. Indeed, we found that Aβ worms expressing CAR-1 K185R are largely (*Figure 4E*) and significantly (*Figure 4F*, from day 7 *p*<0.004) protected from proteotoxicity. The protective effect of CAR-1 K185R is DAF-16-dependent, since no protection was seen when EHC124 worms were treated with *daf-16* RNAi (*Figure 4—figure supplement 1A*). These *daf-16* RNAi-treated worm population exhibited similar rates of paralysis to these of CL2006 worms that were treated with the same RNAi (*Cohen et al., 2006*; *Cohen et al., 2010*). The observation that Aβ worms expressing CAR-1 K257R (strain EHC125) and Aβ worms (CL2006) exhibit indistinguishable rates of paralysis (*Figure 4—figure supplement 1B*) indicates that the counter-proteotoxic effect that is conferred by CAR-1, is suppressed by the SUMOylation of lysine 185. Finally, the more efficient protection from Aβ proteotoxicity that is conferred by *daf-2* RNAi (*Figure 4C*), supports the idea that the protective mechanisms that are activated by IIS reduction and by CAR-1 K185R only partially overlap.

We further examined whether CAR-1 modulates proteotoxicity by employing worms that express fluorescently tagged, poly-glutamine stretches of 67 repeats in their neurons (polyQ67-YFP, strain AM716). Expanded glutamine stretches cause various human neurodegenerative maladies, including Huntington's disease (*Bates, 2003*), and lead to impaired neuronal activity in worms (*Vilchez et al., 2012*). AM716 worms were grown on EV, *daf-2* or *car-1* RNAi bacteria and placed in a drop of liquid at days 1 and 4 of adulthood. To measure proteotoxicity, the number of body bends per 30 s were counted (*Volovik et al., 2014a*). As anticipated, knocking down *daf-2* protected from proteotoxicity at both day 1 (p<0.001) and day 4 (p<0.001) of adulthood. In contrast, *car-1* RNAi treatment decreased the number of body bends at day 4 of adulthood (p<0.001) but not at day 1 (*Figure 5—figure supplement 1*). These results confirm the roles of CAR-1 as a modulator of age-onset proteotoxicity.

## CAR-1 maintains proteostasis through the germline

Our results suggest that CAR-1 modulates proteostasis, at least partially, by negatively regulating *glp-1* activity. To examine this possibility, we used worms that express polyQ35-YFP in their body wall muscles (strain AM140), and thus exhibit a motility defect (*Morley et al., 2002*), and harbor a temperature-sensitive *glp-1* mutant (strain ABZ21). These animals are protected from polyQ-mediated paralysis when grown at 25°C (*Shemesh et al., 2013*). The rates of paralysis of polyQ35-YFP and of ABZ21 worms that were developed at 25°C and were either grown on EV, *daf-2*, or *car-1* RNAi bacteria, were compared. While *car-1* RNAi-treated polyQ35-YFP worms (that express a functional GLP-1) showed an increased rate of paralysis compared to untreated animals, *daf-2* RNAi provided nearly complete protection from paralysis (*Figure 5*, A and B). In contrast, both untreated and *car-1* RNAi-treated ABZ21 worms were protected from paralysis. This set of experiments shows that knocking down *car-1* has no deleterious effect on proteostasis when *glp-1* is inactive ($p < 0.05$), indicating that CAR-1 controls proteostasis through GLP-1.

Since the CAR-1-associated helicase CGH-1, which is expressed in meiotic germ cells, modulates lifespan (*Figure 5—figure supplement 2*, *Supplementary file 5* and (*Seo et al., 2015*)), we tested whether this helicase also controls proteostasis. To address this, we let Aβ worms develop on EV bacteria or treated them with RNAi towards *daf-2*, *car-1*, or *cgh-1*, followed their rates of paralysis and found that the knockdown of *cgh-1* as well as of *car-1*, significantly ($p < 0.035$) increase the rates of paralysis compared to untreated worms (*Figure 5*, C and D). Similar results were obtained when worms that harbor a metastable perlecan (*unc-52 (ts)*), that misfolds and causes paralysis of worms that are grown at 25°C (strain HE250, [*Shemesh et al., 2013*]), were treated with either *car-1* or *cgh-1* RNAi (*Figure 5—figure supplement 3, A and B*). These results further support the view that CAR-1 and CGH-1 promote proteostasis by modulating germ cell activity, plausibly by negatively regulating GLP-1.

## The role of CAR-1 in stress resistance

The roles of CAR-1 in lifespan determination and proteostasis maintenance have led us to ask whether it also controls stress resistance. To examine whether *car-1* influences heat stress resistance we used two worm strains, N2 and CF512 (the exposure of CF512 worms to 25°C during development does not activate the heat shock response [*Volovik et al., 2012*]). The worms were treated from hatching with *car-1* RNAi or left untreated, and exposed at day 1 of adulthood to 35°C for 11 hr. We observed no significant effect of *car-1* RNAi on resistance to heat (*Figure 6*, A and B). Similarly, *car-1* RNAi did not abolish the elevated heat resistance of *daf-2* mutant worms (*Figure 6C*). Surprisingly, the expression of CAR-1 K185R elevated the survival after heat shock compared to control worms (N2, $p < 0.04$), however a trend but not a significant effect was observed when the natural *car-1* was over-expressed (*Figure 6D*).

We next used CF512 worms to analyze resistance to pathogenic bacteria. The worms were treated throughout development with *daf-2*, *daf-16,* or *car-1* RNAi or left untreated (EV). At day 1 of adulthood, the nematodes were transferred onto plates seeded with the pathogenic bacteria *Pseudomonas aeruginosa*. As expected (*Singh and Aballay, 2006*), the knockdown of *daf-2* extended, whereas *daf-16* RNAi shortened the mean survival of the worms compared to control animals (mean survival of $11.96 \pm 0.42$, $4.28 \pm 0.11$ and $5.88 \pm 0.15$ days, respectively, $p < 0.001$). Interestingly, *car-1* RNAi had a small but significant protective effect from pathogenic bacteria (mean survival of $6.77 \pm 0.21$ days, $p < 0.001$, *Figure 6E*, *Supplementary file 6*). To further test this effect, we conducted the reciprocal experiment asking whether the over-expression of the wild-type CAR-1 (strain EHC117) or the K185R CAR-1 (strain EHC118), shortens the survival rates of worms that were cultured on *P. aeruginosa*. Our results (*Figure 6F*) show a small but significant lifespan shortening effect that stemmed from the over-expression of the wild-type ($p < 0.01$) and a non-significant trend in worms that express the K185R CAR-1. These results confirm that CAR-1 is deleterious when the worms are exposed to these pathogenic bacteria.

We also assessed whether CAR-1 is involved in protection from UV radiation by following the survival of CF512 worms that were treated with RNAi as above, and exposed to a sub-lethal dose of UV. The knockdown of *daf-2* protected the worms from UV and the survival rate of the worms treated with *car-1* RNAi was also increased compared to control animals (mean survival rates of $10.45 \pm 0.22$ (*daf-2* RNAi), $9.23 \pm 0.22$ (*car-1* RNAi) and $7.97 \pm 0.21$ (EV) days, respectively, $p < 0.001$)

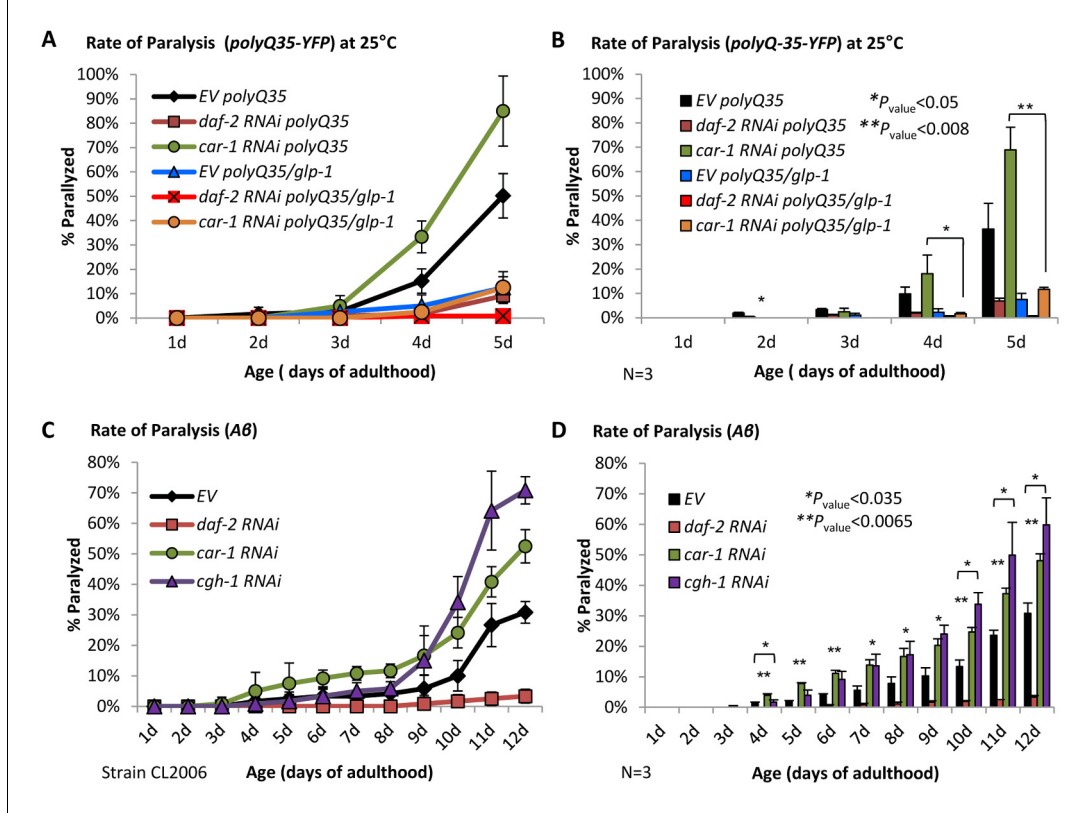

**Figure 5.** CAR-1 modulates proteostasis through the GLP-1 axis. (A–B) Worms expressing polyQ35-YFP in their body wall muscles were crossed with CF1903 animals carrying a ts mutant *glp-1*. PolyQ35-YFP and polyQ35-YFP/*glp-1* worms were exposed during development to 25°C and either left untreated (EV) or treated with *daf-2* or *car-1* RNAi and subjected to paralysis assay. While *daf-2* RNAi protected worms of both strains from paralysis, *car-1* RNAi enhanced paralysis of polyQ35-YFP worms but not of polyQ35-YFP/*glp-1* animals (A). Three independent experiments confirmed the significance of these phenotypes (B). (C–D) Aβ worms were either left untreated or fed with *daf-2, car-1* or *cgh-1* RNAi bacteria and rates of paralysis were followed. While *daf-2* RNAi protected the worms from proteotoxicity, the knockdown of *cgh-1* and *car-1* enhanced paralysis compared to control animals (C). Three independent experiments confirmed the significance of this phenotype (D, p<0.035).
DOI: https://doi.org/10.7554/eLife.38635.018

The following figure supplements are available for figure 5:

**Figure supplement 1.** The number of body bends of worms expressing *polyQ67-YFP* in a pan-neuronal fashion, was measured at days 1 and 4 of adulthood.
DOI: https://doi.org/10.7554/eLife.38635.019

**Figure supplement 2.** The knockdown of *cgh-1* or of *daf-16* by RNAi similarly reduces lifespan of CF512 worms compared to the lifespans of control animals (mean LS of 13.15 ± 0.25, 13.11 ± 0.28 and 16.18 ± 0.40 days respectively, p<0.001).
DOI: https://doi.org/10.7554/eLife.38635.020

**Figure supplement 3.** The knockdown of *car-1* and of *cgh-1* enhance proteotoxicity.
DOI: https://doi.org/10.7554/eLife.38635.021

(*Figure 6G*, *Supplementary file 6*). Similarly, the over-expression of the wild-type CAR-1 or of the CAR-1 K185R mutant significantly shortened the lifespans of worms that were exposed to UV radiation (mean survivals of 12.78 ± 0.33 (EV), 8.89 ± 0.19 (wt CAR-1) and 10.87 ± 0.27 (CAR-1 K185R) days, p<0.001) (*Figure 6H* and *Supplementary file 6*).

Together, these results show that CAR-1 plays minor roles in resistance to heat as well as in survival after exposure to pathogenic bacteria and UV radiation.

The data obtained in this work culminate to suggest the following model (*Figure 7*): besides regulating the cellular localization of its downstream transcription factors (7-I), the IIS also governs aging-associated functions by SUMOylating CAR-1 on lysine 185. This post-translational modification inhibits CAR-1's function (7-II), thereby activating GLP-1 (7-III). Accordingly, the expression of the hyper-active, SUMOylation-resistant CAR-1 K185R, efficiently represses GLP-1 thereby, mimicking

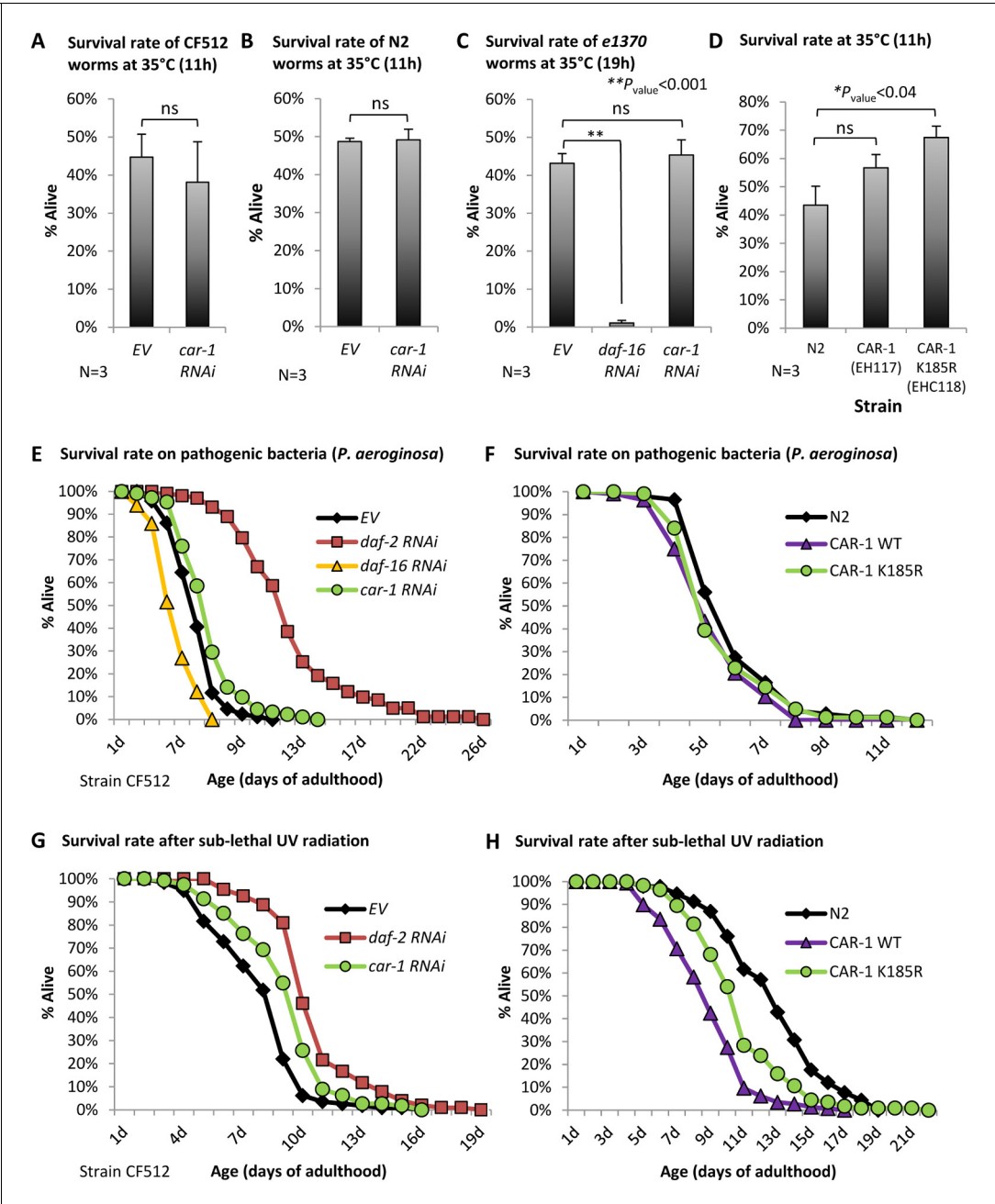

**Figure 6.** The roles of *car-1* in stress resistance. (A–C) The survival rates of heat-stressed CF512 worms (A) N2 animals (B) that were exposed to 35°C for 11 hr, and of *daf-2*(*e1370*) mutant worms (strain CB1370) (C) that were exposed for 19 hr to 35°C, were not significantly affected by *car-1* RNAi. In contrast, *daf-16* RNAi significantly reduced the survival rates of heat-stressed *daf-2* mutant animals (p<0.001). (D) The expression of CAR-1 K185R (EHC118) significantly elevates survival compared to the wild-type animals (average survival of 67.39% and 43.51% respectively, N = 3, p<0.04). A trend but no significant effect was observed in worms that over-express the natural CAR-1 protein (EHC117). (E) *car-1* RNAi-treated CF512 worms are more resistant to the pathogenic bacteria *Pseudomonas aeruginosa* than control worms (mean survival of 6.77 ± 0.21 and 5.88 ± 0.15 days, p<0.001). *daf-2* RNAi prolonged and *daf-16* RNAi reduced the survival of worms that were grown throughout adulthood with *P. aeruginosa* (mean survival rates of 11.96 ± 0.42 and 4.28 ± 0.11 days, respectively, p<0.001 for both treatments). (F) The over expression of wild-type CAR-1 (strain EHC117) or of CAR-1 K185R (strain EHC118) shortens survival of worms that were grown during adulthood on *P. aeruginosa* (mean survival of 6.05 ± 0.17, 5.43 ± 0.14 and 5.66 ± 0.17 days for N2, EHC117 (p<0.005) and EHC118 (p=0.1029), respectively). (G) The survival of CF512 worms that were exposed to sub-lethal dose of UV radiation was significantly increased by *daf-2* and *car-1* RNAi treatments compared to control animals (EV); (mean survival of 10.45 ± 0.22, 9.23 ± 0.22 and 7.97 ± 0.21 days, respectively, p<0.001). (H) In agreement, the over-expression of wild-type CAR-1 (strain EHC117) or of CAR-1 K185R (strain EHC118) shortens survival of animals that were exposed to UV radiation (mean survival of 12.78 ± 0.33, 8.89 ± 0.19 and 10.87 ± 0.27 days for N2, EHC117 (p<0.001) and EHC118 (p<0.001), respectively).

DOI: https://doi.org/10.7554/eLife.38635.022

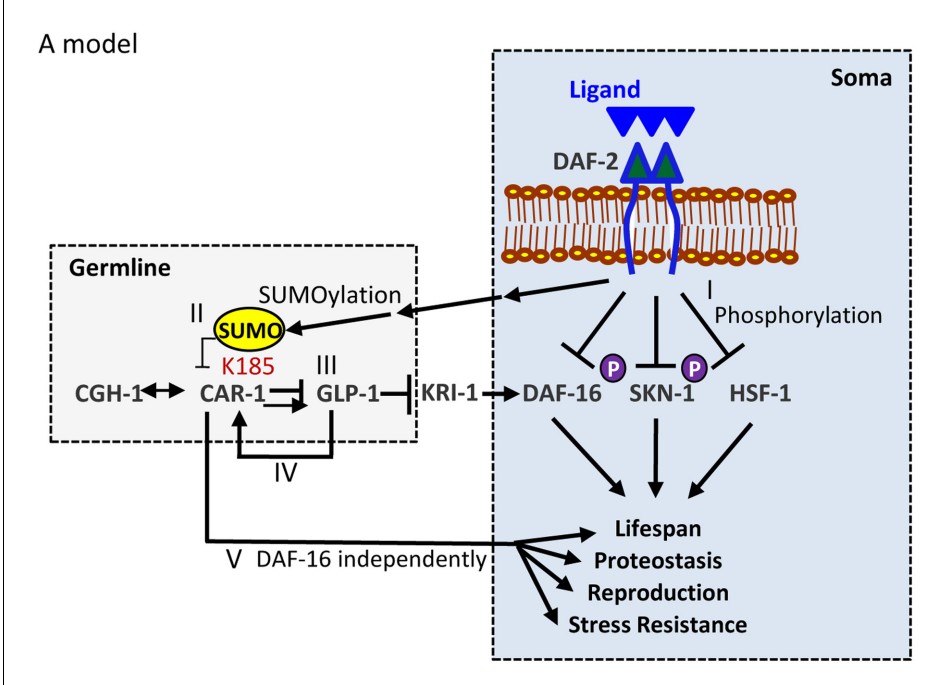

**Figure 7.** A model The IIS negatively regulates its downstream transcription factors DAF-16, SKN-1 and HSF-1 by inhibiting their entrance into the nucleus (I). Thus, knocking down *daf-2* hyper-activates these transcription factors resulting in longevity, proteostasis, stress resistance and modulated reproduction profile. The IIS also governs aging by SUMOylating CAR-1 on lysine 185 to mitigate its regulatory function (II), on GLP-1 (III). The RNA-helicase CGH-1 acts in cooperation with CAR-1 to regulate *glp-1*. IIS reduction hyper-activates CAR-1 by lowering the level of its SUMOylation on K185. This modulates the activity of GLP-1 to mediate longevity and enhance proteostasis in a DAF-16-dependent manner. Our results also indicate that GLP-1 positively controls the expression of *car-1* (IV). CAR-1 also appears to affect lifespan, proteostasis, stress resistance and reproduction by a DAF-16-independent mechanism (**V**).

DOI: https://doi.org/10.7554/eLife.38635.023

one aspect of IIS reduction, and promotes proteostasis. Interestingly, GLP-1 regulates the expression of *car*-1 to create a regulatory circuit (7-IV). Importantly, CAR-1 may also affects lifespan, proteostasis, stress resistance, and reproduction by a DAF-16-independent mechanism (7 V).

## Discussion

Several observations suggest a mechanistic link between the IIS and the germline. First, knocking down *daf-2* extends lifespan solely during reproductive adulthood (*Dillin et al., 2002*). This finding is particularly interesting as other functions of the IIS are regulated at other stages of the nematode's lifecycle; egg laying is controlled during development (*Dillin et al., 2002*) and proteostasis is governed by the IIS during early and late adulthood (*Cohen et al., 2010*). Moreover, the key roles of DAF-16 in the longevity phenotype that results from both, IIS reduction (*Kenyon et al., 1993*) and loss of germ cells (*Berman and Kenyon, 2006*), further support the idea that the IIS and germline are inter-related. Indeed, DAF-2 signaling was reported to govern oogenesis through the RAS-ERK pathway according to food availability (*Lopez et al., 2013*) and the phosphatase DAF-18/PTEN locally antagonizes IIS activity in the germline (*Narbonne et al., 2015*). In addition, germline signaling synergizes the longevity mechanisms downstream of the IIS and of the target of rapamycin (TOR) pathway (*Chen et al., 2013*). Nevertheless, whether the IIS and the germline coordinate aging and proteostasis in an orchestrated manner and if so, how this coordination is achieved, remained largely unexplored. Here, we show that IIS reduction lessens the SUMOylation of CAR-1, a germline protein that is involved in mRNA processing and negatively regulates the levels of *glp-1* (*Noble et al., 2008*). Our data suggest that by SUMOylating CAR-1, the IIS suppresses its activity, thereby

activating *glp-1* to shorten lifespan and impair proteostasis. This notion is supported by several observations. First, the knockdown of *car-1* shortens the lifespans of CF512 and of *daf-2* mutant worms (*Figure 2B* and *Figure 2—figure supplement 1, A–B*), but not of animals that lack functional *glp-1* or *kri-1* (*Figure 3*, A and B). On the other hand, the knockdown of *car-1* exhibits similar shortening effects on the lifespans of CF512 and *daf-2* mutant animals, questioning the role of *car-1* in the lifespan-controlling functions of the IIS. Nevertheless, the expression of CAR-1 K185R which lacks a putative SUMOylation site, extends lifespan (*Figure 2*, D and E) and protects model worms from proteotoxicity (*Figure 4*, E and F) (it is important to note that it is not clear whether this mutated CAR-1 restores the natural functions of the protein or activate another lifespan extending mechanism). In addition, the simultaneous knockdown of *daf-2* and *car-1* prevents IIS reduction from conferring its full protective effect on Aβ worms (*Figure 4*, C and D). These results argue that CAR-1 is needed for IIS reduction to fully protect the worm from proteotoxicity. Finally, using *daf*-2 mutant worms and qPCR we found that the knockdown of *car-1* by RNAi elevates the expression of *sygl-1* and lessens the levels of *lst-1* (*Figure 3I*), which are regulated by GLP-1 (*Figure 3—figure supplement 3, A and B*). These results indicate that CAR-1 is a co-regulator of GLP-1 activity, however, while in the case of *sygl-1* the knockdown of *car-1* activates GLP-1-mediated transcription, *car-1* RNAi treatment reduces the expression of *lst-1.* These results show that CAR-1 can function as either negative or positive regulator of GLP-1 and show that the relations between CAR-1 and GLP-1 require further elucidation.

## CAR-1 and SUMOylation coordinate signaling of the IIS and the germline but probably affect lifespan through an additional mechanism

Post-translational modifications are known to have various biological functions, including the regulation of aging. Phosphorylation regulates the activities of DAF-16, SKN-1, and HSF-1 downstream of the IIS (*Lee et al., 2001*; *Tullet et al., 2008*; *Chiang et al., 2012*) and SUMOylation controls the localization of the IGF-1 receptor and its signaling activity in mammalian tissues (*Sehat et al., 2010*). SUMOylation is also critical for aging-associated modulation of mevalonate biosynthesis (*Sapir et al., 2014*), a metabolite that has been implicated in the development of clinical conditions (*Mokarram et al., 2017*). In this study, we unveiled a novel role of SUMOylation in the regulation of aging, serving as a functional switch of CAR-1, which is governed by the IIS. This raises the question of how the IIS controls the SUMOylation state of CAR-1. One possible explanation stems from the correlation between AKT function, the stability of SUMO, and the SUMO-conjugating enzyme UBC-9 in mammals (*Lin et al., 2016*). According to this theme, reduced IIS lowers AKT activity, resulting in SMO-1 destabilization and in reduction of UBC-9 activity. This cascade of events may lower the rate of global protein SUMOylation. This possibility appears less likely as our results (*Figure 1A*) indicate that while *daf-2* RNAi leads to lower SUMOylation of some proteins others proteins exhibit increased SUMOylation upon IIS reduction. Alternatively, the expression of specific genes that encode for proteins involved in CAR-1 SUMOylation may be positively regulated by the IIS, which lowers the expression of these genes and reduces the rate of CAR-1 SUMOylation. Future research is needed to clarify this issue.

Although SUMOylation appears to be a pivotal post-translational modification that influences aging, a simultaneous knockdown of *car-1* and *daf-2* only partially protects worms from proteotoxicity and only partially shortens lifespans of *daf-2* mutant worms. On one hand, these observations suggest that IIS reduction also protects from proteotoxicity and extends lifespan by additional, CAR-1-independent mechanisms. However, on the other hand, the knockdown of *daf-16* in CAR-1 K185R-expressing worms did not shorten lifespan below those of wild-type nematodes (*Figure 2G*), suggesting that CAR-1 also governs lifespan by an additional, DAF-16-independent mechanism (the lack of functional *daf-16* shorten the lifespan of N2 worms by approximately 30% [*Kenyon et al., 1993*]). Nevertheless, the knockdown of *car-1* has not further shortened the lifespan of *daf-16* mutant animals (*Figure 2C*). The lack of additive effect may be a result of the very short lifespan of *daf-16* mutant worms, which does not allow the knockdown of *car-1* to further shorten lifespan or perhaps by the limited efficiency of RNAi-mediated knockdown of *car-1*.

## Opposing effects of car-1 on stress resistance and proteostasis

An additional interesting aspect of this study is the differential effects of *car-1* on distinct environmental insults. While the knockdown of *car-1* has no effect on heat stress resistance (*Figure 6*, A-C), the expression of CAR-1 K185R, mildly but significantly elevates the survival of heat-stressed worms (*Figure 6D*). In contrast, the knockdown of *car-1* has a small but reproducible protective effect on resistance to UV radiation (*Figure 6G*) and to pathogenic bacteria (*Figure 6E*). In agreement, the over expression of either the wild-type CAR-1 or the mutated CAR-1 K185R is deleterious to worms that were exposed to these insults (*Figure 6*, F and H). The observed protection from UV radiation, conferred by *car-1* RNAi, is consistent with a previous report that CAR-1 and CGH-1 negatively regulate DNA-damage-mediated apoptosis (*Tomazella et al., 2012*). Nevertheless, despite its protective effect when the worm is exposed to stress conditions, the knockdown of *car-1* shortens lifespan (*Figure 2B*). These results support the theme that the ability to resist stresses such as heat (*Maman et al., 2013*; *Volovik et al., 2014b*) and oxidation (*Van Raamsdonk and Hekimi, 2012*) are not necessarily coupled with lifespan determination. They also coincide with the reports that IIS-regulated factors may be involved in the regulation of certain stress resistance mechanisms but not of others. For instance, the transcription factor SMK-1 is needed for the worm to resist UV radiation and pathogenic bacteria but is dispensable for coping with heat (*Wolff et al., 2006*). Similarly, we recently reported that the knockdown of caveolin-1 extends lifespan and provides partial protection from pathogenic bacteria, but has no role in heat stress resistance (*Roitenberg et al., 2018*).

The *car-1* RNAi-mediated protection from certain stresses appears to be contradictive to the observation that knocking down this gene exposes the animal to proteotoxicity. However, it has been already shown that abolishing the nematode's ability to resist heat by knocking down neuronal components, provides the worm with partial protection from proteotoxicity (*Prahlad and Morimoto, 2011*; *Volovik et al., 2014b*). Our findings show that manipulating the activity of a germline protein can also confer opposing effects on stress resistance and proteostasis, and raise the question of how CAR-1 promotes these opposing effects. One possible explanation suggests that CAR-1 may differentially affect the expression levels of different GLP-1-controlled genes. Such differential effects of transcriptional co-regulators have been reported. For instance, the knockdown of the DAF-16 transcriptional co-factor *nhl-1,* lowers the expression level of *sod-3* and of *sip-1*, but has no effect on the expression of *mtl-1*, which are all known target genes of DAF-16 (*Volovik et al., 2014b*). The opposing effects of *car-1* RNAi on the expression levels of *sygl-1* and *lst-1* propose a similar mechanism of differential effects on the expression of distinct genes. How this mechanism functions and what cellular components are involved in the mediation of proteostasis by the SUMOylation-resistant CAR-1, are questions that require further elucidation.

## Materials and methods

### Key resources table

| Reagent type (species) or resource | Designation | Source or reference | Identifiers | Additional information |
|---|---|---|---|---|
| Strain, strain background (*Caenorhabditis elegans*) | N2 | *Caenorhabditis* Genetic Center (CGC) | https://cgc.umn.edu/strain/N2 | |
| Strain, strain background (*Caenorhabditis elegans*) | CF512 | *Caenorhabditis* Genetic Center (CGC) | https://cgc.umn.edu/strain/CF512 | |
| Strain, strain background (*Caenorhabditis elegans*) | CF1903 | *Caenorhabditis* Genetic Center (CGC) | https://cgc.umn.edu/strain/CF1903 | |
| Strain (*Caenorhabditis elegans*) | CB1370(e1370) | *Caenorhabditis* Genetic Center (CGC) | https://cgc.umn.edu/strain/CB1370 | |
| Strain, strain background (*Caenorhabditis elegans*) | e1368 | | | Dr. Andrew Dillin, University of California, Berkeley, USA |

*Continued on next page*

*Continued*

| Reagent type (species) or resource | Designation | Source or reference | Identifiers | Additional information |
|---|---|---|---|---|
| Strain, strain background (*Caenorhabditis elegans*) | CF1038 | *Caenorhabditis* Genetic Center (CGC) | https://cgc.umn.edu/strain/CF1038 | |
| Strain, strain background (*Caenorhabditis elegans*) | CF2052 | *Caenorhabditis* Genetic Center (CGC) | https://cgc.umn.edu/strain/CF2052 | |
| Strain, strain background (*Caenorhabditis elegans*) | AA86 | *Caenorhabditis* Genetic Center (CGC) | https://cgc.umn.edu/strain/AA86 | |
| Strain, strain background (*Caenorhabditis elegans*) | CF2531 | *Caenorhabditis* Genetic Center (CGC) | | |
| Strain, strain background (*Caenorhabditis elegans*) | ABZ21 | other | | Dr. Anat Ben-Zvi, Ben Gurion University, Israel |
| Strain, strain background (*Caenorhabditis elegans*) | CL2006 | *Caenorhabditis* Genetic Center (CGC) | https://cgc.umn.edu/strain/CL2006 | |
| Strain, strain background (*Caenorhabditis elegans*) | WH377 | *Caenorhabditis* Genetic Center (CGC) | https://cgc.umn.edu/strain/WH377 | |
| Strain, strain background (*Caenorhabditis elegans*) | WH346 | *Caenorhabditis* Genetic Center (CGC) | https://cgc.umn.edu/strain/WH346 | |
| Strain, strain background (*Caenorhabditis elegans*) | NX25 | other | | Dr. Limor Broday, Tel-aviv University, Israel |
| Strain, strain background (*Caenorhabditis elegans*) | AM140 | *Caenorhabditis* Genetic Center (CGC) | https://cgc.umn.edu/strain/AM140 | |
| Strain, strain background (*Caenorhabditis elegans*) | AM716 | other | | Dr. Richard I Morimoto, Northwestern University, IL, USA |
| Strain, strain background (*Caenorhabditis elegans*) | HE250 | *Caenorhabditis* Genetic Center (CGC) | https://cgc.umn.edu/strain/HE250 | |
| Strain, strain background (*Caenorhabditis elegans*) | EHC117 | This paper | N/A | pUC18-car-1p::2HAcar-1 injected to N2 worms |
| Strain, strain background (*Caenorhabditis elegans*) | EHC118 | This paper | N/A | pUC18-car-1p::2 HA car-1 K185R injected to N2 worms |
| Strain, strain background (*Caenorhabditis elegans*) | EHC121 | This paper | N/A | pUC18-car-1p::2 HA car-1 K257R injected to N2 worms |
| Strain, strain background (*Caenorhabditis elegans*) | EHC124 | This paper | N/A | pUC18-car-1p::2 HA car-1 K185R injected to CL2006 worms |
| Strain, strain background (*Caenorhabditis elegans*) | EHC125 | This paper | N/A | pUC18-car-1p::2 HA car-1 K257R injected to CL2006 worms |
| Genetic reagent | empty vector (EV) (pAD12) | DOI: 10.1126/science.1074240 | ID_addgene: 34832 | |
| Genetic reagent | *daf-2* RNAi (pAD48) | DOI: 10.1126/science.1074240 | ID_addgene: 34834 | |
| Genetic reagent | *daf-16* RNAi (pAD43) | DOI: 10.1126/science.1074240 | ID_addgene: 34833 | |

*Continued on next page*

*Continued*

| Reagent type (species) or resource | Designation | Source or reference | Identifiers | Additional information |
|---|---|---|---|---|
| Genetic reagent | *car-1* RNAi | vidal RNAi library | Product code: 3320_Cel_ORF_RNAi | |
| Genetic reagent | *cgh-1* RNAi | vidal RNAi library | Product code: 3320_Cel_ORF_RNAi | |
| Antibody | anti-GFP antibody (Rabbit monoclonal) | Cell Signaling | ca#2956 | (1:1000) |
| Antibody | anti-SUMO-1 antibody (Rabbit polyclonal) | Millipore | ca#09–409 | (1:2000) |
| Antibody | anti-HA.11 epitope tag (Mouse monoclonal) | BioLegend | ca#901501 | (1:2000) |
| Antibody | anti-FLAG M2, Clone M2 (Mouse monoclonal) | Sigma | ca#F1804 | (1:1000) |
| Antibody | anti-actin antibody (Mouse monoclonal) | Sigma | ca#A5441 | (1:5000) |
| Commercial assay or kit | HisPur$^{TM}$ Ni-NTA Resin | Thermo Fisher Scientific | ca#88221 | |
| Commercial assay or kit | Red ANTI-FLAG M2 Affinity Gel | Sigma | ca#F2426 | |
| Commercial assay or kit | GFP-Trap_A | Chromotek | code#gta-100 | |
| Commercial assay or kit | Pierce Crosslink Immunoprecipitation Kit | Thermo Fisher Scientific | ca#26147 | |
| Commercial assay or kit | NucleoSpin RNA kit | MACHEREY-NAGEL | ca#740955.50 | |
| Commercial assay or kit | iScript cDNA Synthesis Kit | Biorad | ca#170–8891 | |
| Commercial assay or kit | EvaGreen supermix | Biorad | ca#172–5204 | |
| Commercial assay or kit | BCA kit | Thermo Fisher Scientific | ca#23225 | |
| Sequence-based reagent | qPCR act-1 forward primer (5'–>3') | This paper, IDT | N/A | GAG CAC GGT ATC GTC ACC AA |
| Sequence-based reagent | qPCR act-1 reverse primer (5'–>3') | This paper, IDT | N/A | TGT GAT GCC AGA TCT TCT CCA T |
| Sequence-based reagent | qPCR cdc-42 forward primer (5'–>3') | This paper, IDT | N/A | CTG CTG GAC AGG AAG ATT ACG |
| Sequence-based reagent | qPCR cdc-42 reverse primer (5'–>3') | This paper, IDT | N/A | CTC GGA CAT TCT CGA ATG AAG |
| Sequence-based reagent | qPCR car-1 forward primer (5'–>3') | This paper, IDT | N/A | AGG AGA GAG AAA CGA ATC AG |
| Sequence-based reagent | qPCR car-1 reverse primer (5'–>3') | This paper, IDT | N/A | TTG TAA CCT CCA TAT CCG C |
| Sequence-based reagent | qPCR sygl-1 forward primer (5'–>3') | This paper, IDT | N/A | AGG CAA AGG AAT CAA GC |
| Sequence-based reagent | qPCR sygl-1 reverse primer (5'–>3') | This paper, IDT | N/A | TTA CGA TAC TTC AGG TTG G |
| Sequence-based reagent | qPCR lst-1 forward primer (5'–>3') | This paper, IDT | N/A | CCA CGC TTG TTA TTT TCG |
| Sequence-based reagent | qPCR lst-l-1 reverse primer (5'–>3') | This paper, IDT | N/A | AGT TGT TTC TTC TTG GAG G |

*Continued on next page*

*Continued*

| Reagent type (species) or resource | Designation | Source or reference | Identifiers | Additional information |
|---|---|---|---|---|
| Software, algorithm | Mass Spectrometry | The PRIDE PRoteomics IDEntifications (PRIDE) database | ID_pride archive: PXD010011 | |
| Software, algorithm | Computational tool GPS-SUMO | DOI: 10.1093/nar/gku383 | | |
| Software, algorithm | ImageJ | NIH | https://imagej.nih.gov/ij/ | |

## Worm and RNAi strains

N2 (wild-type, Bristol), CB1370 (*daf-2(e1370)* mutant worms), CL2006 (*unc-54p::human Aβ₃₋₄₂*), CF512 (*fer-15*(b26)II; *fem-1*(hc17)IV), CF1903 (*glp-1(e2141)* III.), AM140 (*Punc54::Q35::YFP*), HE250 (*unc-52(e669su250)* II.), WH377 (*car-1(tm1753)* I/hT2), WH346 (*unc-119*(ed3) III. ojIs34 [*GFP::car-1 +unc-119(+)*], CF2052 kri-1(ok1251) (I), AA86 *daf-12*(rh61rh411) X., CF2531 *daf-9*(rh50) X., CF1038 (*daf-16*(mu86)I) were obtained from the *Caenorhabditis* Genetic Center (CGC, Minneapolis, MN). Worms that over-express the K185R or K257R mutated *car-1* (strains EHC118 and EHC121 respectively) or WT CAR-1 tagged to 2xHA tag (strain EHC117) were generated by injecting a plasmid that carries the gene downstream of the natural *car-1* promoter region (1074 bp upstream of the ORF) into N2 worms. *rol-6* driven by the *unc-54* promoter or *gfp* driven by the *elt-2* promoter were used as selection markers. Worms expressing *Aβ* in their body wall muscle and CAR-1 K185R or CAR-1 K257R were generated by injecting the same plasmids into CL2006 worms (strains EHC124 and EHC125, respectively). AM716 (*rmIs284[pF25B3.3::Q67::YFP]*) worms were obtained from Dr. Richard I Morimoto (Northwestern, IL). ABZ21 animals (*Punc54::Q35::YFPxglp-1* (CF1903)) were a gift of Dr. Anat Ben-Zvi (Ben-Gurion, Israel). NX25 (*smo-1* (ok359);tvEx25[*psmo-1::His-FLAG-SMO-1; rol-6*]) were obtained from Dr. Limor Broday (TAU, Israel). CF512 (*fer-15*(b26)II; *fem-1* (hc17)IV), CF1903 (*glp-1*(e2144) III.) nematodes are heat-sensitive sterile and were thus, grown at 15°C. To avoid egg lying, these worms were developed at 25°C and transferred at day 1°C to 20°C until harvested. To achieve sterility, ABZ21 worms were grown at 25°C until harvesting. Other strains were synchronized and grown on the indicated RNAi bacteria at 20°C until day 1 of adulthood. To reduce gene expression, we used bacterial strains expressing dsRNA: empty vector (pAD12), *daf-2* (pAD48), *daf-16* (pAD43). *car-1* and *cgh-1* dsRNA-expressing bacteria were obtained from the Vidal RNAi library. RNAi bacteria were grown at 37°C in LB with 100 μg/ml ampicillin and then seeded on NG-ampicillin plates with the addition of 100 mM Isopropyl β-D-1-thiogalactopyranoside (IPTG ~ 1 mM final concentration).

## Purification of SUMOylated proteins

To isolate SUMOylated proteins, 300,000 NX25 worms were grown on EV or *daf-2* RNAi bacteria until day 1 of adulthood, collected and froze in liquid nitrogen (*Figure 1*, B, C and D). The worms were then homogenized and equilibration buffer (1XPBS, 8M UREA, 5 mM NEM and protease inhibitors) was added prior to centrifugation (10 min, 9,391 g). Protein concentrations were measured and equalized by Bradford reagent. First, a His-tag purification using HisPur Ni-NTA Resin (Thermo Scientific, #88221) was performed. Lysates were incubated with the resin for 50 min at RT and washed (1XPBS (pH7.4), 8M UREA, 250 mM Imidazole). SUMOylated proteins were eluted using buffer 1 (1XPBS, 2M UREA, 250 mM Imidazole) followed by elution buffer 2 (1XPBS, 1M UREA, 250 mM Imidazole). Aliquots of the samples were blotted by WB for validation. The remaining eluted samples were used for Flag-tag purification with the Red ANTI-FLAG M2 Affinity Gel (Sigma-Aldrich, #F2426). Samples were diluted with RIPA buffer (50 mM Tris HCl pH7.5, 150 mM NaCl, 5 mM EDTA, 1% TritionX-100, 0.1% SDS, 1X Protease Inhibitor (Calbiochem set III #539134, 1 mM BetaME)) and incubated over night at 4°C with the Anti-Flag beads. The beads were washed with RIPA buffer followed by elution (100 mM Glycine pH3.5, 150 mM NaCl).

To detect the SUMOylation state of CAR-1 (*Figure 1*, C and D), WH346 worms were treated as described above. The homogenized worms were dissolved in RIPA buffer and the GFP immunoprecipitation was performed using GFP-Trap_A (#gta-100, Chromotek, Germany) according to the

manufacturer's instructions. The beads were incubated with the lysates over night at 4°C and the trapped proteins were eluted and analyzed by WB.

To test the SUMOylation state of the CAR-1 (WT), CAR-1 K185R and CAR-1 K257R mutants (strains EHC117, EHC118 and EHC121, respectively) 120,000 worms were harvested as described above. CAR-1 was purified by performing an immunoprecipitation using the anti-HA.11 Epitope Tag antibody and the Pierce Crosslink Immunoprecipitation Kit. The crosslinked beads and the worm lysates were incubated over night at 4°C, bound proteins were eluted and analyzed by WB.

## RNA isolation and quantitative real-time PCR

Total RNA was isolated from synchronized worm populations using QIAzol reagent (QIAGEN, Hilden Germany #79306) and NucleoSpin RNA kit (MACHEREY-NAGEL, #740955.50). cDNA was synthesized using iScript cDNA Synthesis Kit (Biorad, #170–8891). Quantitative real-time PCR reactions were performed with EvaGreen supermix (Biorad, #172–5204). Quantities were normalized to levels of *act-1* and of *cdc-42* cDNA.

## SDS-PAGE and western blot analysis

To blot SUMOylated proteins (*Figure 1A*), N2, CF512, and CF1903 worms that were grown at 15°C, were bleached to obtain synchronized eggs. The eggs were placed on plates that were seeded with control bacteria (EV) or *daf-2* RNAi bacteria and incubated for 48 hr at 25°C (to sterilize the CF512 and CF1903 worms). The worms were transferred thereafter to 20°C for additional 24 hr. For the experiment displayed at *Figure 1E*, the worms were hatched and grown at 20°C (strain WH346). At day 1 of adulthood, the worms were washed twice with M9, and homogenized using a bullet grinder (full speed, 10 s, three times). The worm homogenates were spun for 3 min at 850 g (3000 rpm in a benchtop Qiagen centrifuge) to sediment debris. The post debris supernatants were collected, protein amounts were measured by a BCA kit (Thermo Fisher #23225), supplemented with loading buffer (10% glycerol, 125 mM Tris base, 1% SDS) and heated at 95°C for 10 min. For each treatment, equal protein quantities were loaded and separated by 10% sodium dodecyl sulfate polyacrylamide gel electrophoresis (SDS-PAGE), transferred onto a PVDF membrane (Millipore, Billerica MA) and probed with the indicated antibody: GFP antibody (Cell Signaling, Danvers, MA cat #2956), anti-SUMO-1 antibody (Millipore, #09–409), anti-HA.11 antibody (BioLegend, San Diego, CA, #901501) or anti-actin antibody (Simga, #A5441). HRP-conjugated secondary antibody and a luminescent image analyzer (ChemiDoc XRS + BioRad) were used to detect protein signals.

## Lifespan and paralysis assays

Synchronized worm eggs were placed on master NG-Ampicillin plates seeded with the indicated RNAi bacterial strain and supplemented with 100 mM IPTG. The eggs were incubated at 20°C until transferred onto small NG- Ampicillin plates, 12 animals per plate (CF1903 and CF512 were incubated throughout development at 25°C, to induce sterility). Adult worms were transferred onto freshly seeded plates every 3 days. Worms that failed to move their noses when tapped twice with a platinum wire were scored as dead. Dead worms were scored daily. Lifespan analyses were conducted at 20°C.

## Heat, UV and innate immunity stress assays

For all stress assays synchronized eggs were placed on NG plates seeded with the RNAi bacteria (as indicated). For heat-stress assays, 120 day one adult animals were transferred onto fresh plates (12 animals per plate) spotted with RNAi bacteria and exposed to 35°C (N2, EHC117, EHC118, EHC121 and CF512 worms for 11 hr and CB1370 worms for 19 hr) and survival rates were recorded. To assess resistance to ultra-violet (UV) radiation, day 1 adult CF512 worms were exposed to sub-lethal UV dose (800 j/cm$^2$). Survival rates were scored daily. To evaluate resistance to pathogenic bacteria (innate immunity), eggs of *CF512* worms were placed on plates seeded with the indicated RNAi bacteria, grown to day 1 of adulthood, and transferred onto plates seeded with *P. aeruginosa*. Survival rates were followed daily.

## Germ cells number quantification

20–24 hr post L4 worms were dissected in egg buffer (0.025 mM Hepes pH 7.4, 118 mM NaCl, 48 mM KCl, 2 mM $MgCl_2$, 2 mM $CaCl_2$, 0.1% Tween 20), transferred to superfrost plus slide and freeze cracked. Gonads were fixed in −20°C MeOH for 1 min, and 4% PFA for 30 min. Slides were washed twice in PBST (PBS with 0.1% Tween 20) for 5 min, and incubated in PBST with 0.5 µg/ml DAPI for 10'. Finally, the slides were washed for 10 min in PBST, again in 10 mM Tris 7.5% and 0.1% Tween 20 for 5 min, and sealed with Vectashield (Vector Laboratories, # H-1000). Imaging was done with Olympus IX81 inverted fluorescent microscope, and 3D images we collected and deconvolved with AutoQuant X3. Germ cells nuclei were manually counted from the mitotic tip to the end of pachytene.

## Quantitative analysis of germ-cell apoptosis

Germ cell corpses were scored in 20 hr post-L4 adult hermaphrodites using acridine orange (AO), as described in *Melchior (2000)*. A minimum of 23 gonads were scored for each genotype. Statistical analyses were performed using the two-tailed Mann–Whitney test (95% C.I.)

## Statistical analyses

Statistical significance of the results was performed using the Student T-test, two-tailed distribution and two-sample equal variance. The analyses were done using at least three independent biological repeats of each experiment, as indicated. Statistical information of lifespan experiments is presented in *Supplementary files 2–6* as mean LS ± SEM.

## Acknowledgements

This study was supported by the European Research Council (ERC) (EC#281010) and the Israel Science Foundation (ISF) EC#981/16 and YBT#1283/15 and 2090/15.

## Additional information

### Funding

| Funder | Grant reference number | Author |
|---|---|---|
| Israel Science Foundation | EC 981/16 | Hana Boocholez<br>Danielle Grushko<br>Ehud Cohen |
| Israel Science Foundation | YBT 2090/15 | Bar Lampert<br>Yonatan B Tzur |
| Israel Science Foundation | YBT 1283/15 | Bar Lampert<br>Yonatan B Tzur |
| European Research Council | EC 281010 | Lorna Moll<br>Noa Roitenberg<br>Michal Bejerano-Sagie<br>Filipa Carvalhal Marques<br>Yuli Volovik<br>Tayir Elami<br>Danielle Grushko<br>Ehud Cohen |

The funders had no role in study design, data collection and interpretation, or the decision to submit the work for publication.

### Author contributions

Lorna Moll, Conceptualization, Data curation, Validation, Investigation, Methodology, Writing—review and editing; Noa Roitenberg, Conceptualization, Data curation, Investigation, Methodology, Writing—review and editing; Michal Bejerano-Sagie, Conceptualization, Validation, Investigation, Writing—original draft; Hana Boocholez, Investigation, Performed stress resistance assays including heat, UV and survival on pathogenic bacteria as well as lifespan assays, Involved in data analysis and

interpretation; Filipa Carvalhal Marques, Investigation, Performed paralysis assays and created transgenic worms, Assisted with data interpretation; Yuli Volovik, Investigation, Performed paralysis assays; Tayir Elami, Investigation, Performed UV stress assays and paralysis assays; Atif Ahmed Siddiqui, Methodology, Created transgenic worms for the project; Danielle Grushko, Investigation, Performed qPCR experiments; Adi Biram, Investigation, Conducted Western blot experiments; Bar Lampert, Investigation, Performed germ cell analyses in worm gonads; Hana Achache, Methodology, Compared the number of apoptotic nuclei in worm gonads; Tommer Ravid, Yonatan B Tzur, Investigation, Writing—review and editing; Ehud Cohen, Conceptualization, Resources, Formal analysis, Supervision, Funding acquisition, Validation, Methodology, Writing—original draft, Project administration

### Author ORCIDs
Noa Roitenberg (iD) http://orcid.org/0000-0002-2181-3313
Adi Biram (iD) https://orcid.org/0000-0001-6169-9861
Ehud Cohen (iD) http://orcid.org/0000-0001-5552-7086

### Decision letter and Author response
Decision letter https://doi.org/10.7554/eLife.38635.034
Author response https://doi.org/10.7554/eLife.38635.035

## Additional files

### Supplementary files
• Supplementary file 1. Differentially SUMOylated proteins (related to *Figure 1*)
DOI: https://doi.org/10.7554/eLife.38635.024

• Supplementary file 2. Lifespans of different strains that were grown on *daf-16* or *car-1* RNAi (related to *Figures 2* and *3*)
DOI: https://doi.org/10.7554/eLife.38635.025

• Supplementary file 3. Lifespans of different worm strains treated with *daf-16* or *car-1* RNAi (related to *Figures 2* and *3*)
DOI: https://doi.org/10.7554/eLife.38635.026

• Supplementary file 4. Lifespans of different worm strains treated with *daf-16* RNAi or *car-1* RNAi (related to *Figure 4*)
DOI: https://doi.org/10.7554/eLife.38635.027

• Supplementary file 5. Survival of CF512 worms on *cgh-1* RNAi (related to *Figure 6*)
DOI: https://doi.org/10.7554/eLife.38635.028

• Supplementary file 6. Survival of CF512 worms on pathogenic bacteria or after UV radiation (related to *Figure 6*)
DOI: https://doi.org/10.7554/eLife.38635.029

• Transparent reporting form
DOI: https://doi.org/10.7554/eLife.38635.030

### Data availability
Mass Spectrometry data has been deposited at http://www.ebi.ac.uk/pride (Project accession: PXD010011).

The following dataset was generated:

| Author(s) | Year | Dataset title | Dataset URL | Database and Identifier |
|---|---|---|---|---|
| Moll L, Roitenberg N, Bajerano-Sagie M, Boocholez H, Carvalhal Marques F, Volovik Y, Elami T, Ahmed Siddiqui A, Grushko D, Bir- | 2018 | The Insulin/IGF Signaling Cascade Modulates SUMOylation to Regulate Aging and Proteostasis in C. elegans | https://www.ebi.ac.uk/pride/archive/projects/PXD010011 | EBI PRIDE, PXD010011 |

am A, Lampert B,
Achache H, Ravid T,
Tzur YB, Cohen E

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
