## [Decision Letter]

Thank you for submitting your article "The Insulin/IGF Signaling Cascade Modulates SUMOylation to Regulate Aging and Proteostasis in *C. elegans*" for consideration by *eLife*. Your article has been reviewed by K VijayRaghavan as the Senior Editor, a Reviewing Editor, and three reviewers. The following individuals involved in review of your submission have agreed to reveal their identity: Veena Prahlad (Reviewer #1); Andrew Dillin (Reviewer #2). A further reviewer remains anonymous.

The reviewers have discussed the reviews with one another and the Reviewing Editor has drafted this decision to help you prepare a revised submission.

Summary:

The manuscript "The Insulin/IGF Signaling Cascade Modulates SUMOylation to Regulate Aging and Proteostasis in *C. elegans*" by Moll et al., presents a very detailed description of one of the molecular mechanisms downstream of IIS-mediated proteotoxicity and lifespan extension. The Cohen group have done an extraordinary amount of work highlighting the molecular mechanism of CAR-1, which is described as a protein downstream of the IIS cascade and upstream of GLP-1 in regulation of this highly studied, but still poorly understood, paradigm of lifespan extension. The experimental methods are elegant and there is a lot of data presented, but the style of the data presentation and writing invite a lot of questions and potential experiments, all of which are listed below. Most of the comments can be addressed textually, and many of the experiments are perhaps not necessary should the writing be made more concise and informative. Again, a majority of the experimental suggestions and questions came up due to the presentation style, which had a major lack of descriptive reasoning behind the experiments, the methods by which the experiments were performed, and conclusions drawn from the data. Either the text needs to be cleaned up considerably to remove the confusions drawn out below, or perhaps the experiments suggested need to actually be performed.

Essential revisions:

1) Is the increased sumoylation found in CF512 (Figure 1A) due to differences that result in growth at 25 °C? Were the N2 worms also grown at 25 °C for this assay to control for this difference? The Materials and methods do not make it clear – it looks like it was performed at different temperatures. Several studies have shown that 25 °C can cause dramatic differences in proteostasis, so this needs to be re-evaluated or made clearer in the text. The authors are also urged to check other reproduction-deficient mutants, such as glp-1 and glp-4, or via chemical sterilization, such as FUDR, to see if these major differences are due to the lack of germline, or specific to CF512. If it is specific to CF512 and the reasoning behind why there are such major differences in sumoylation cannot be rectified, this begs the question of whether this is an appropriate model to use. Another question that comes up is whether CAR-1 protein itself is differentially sumoylated in CF512 background.

2) Figure 1C-D are confusing. How do we know that 1C is showing SUMOylated-GFP-CAR-1 and not other protein that gets pulled down with an anti-SUMO antibody? Is this actually a double-IP where it is pulled down with SUMO antibodies, then GFP antibodies? This is especially confusing if probing with GFP-antibodies cannot visualize the larger, SUMOylated versions of the proteins – does SUMOylation prevent the GFP antibody from binding? Please clear up the experimental details and the conclusions drawn from the data.

3) Since car-1 RNAi/KO decreases lifespan of wild-type and daf-2 mutants to a similar extent, is this just an additive phenotype? (Figure 2A-B)

4) A single copy rescue experiment of the CAR-1 K185R should be performed in the car-1 KO to ensure that this CAR-1 still maintains the normal (albeit hyperactive) functions of the protein (e.g. rescues sterility) and isn't a completely mutant form of the protein with brand new functions.

5) The lifespan decrease of car-1 RNAi on daf-9 mutants seems much milder than in wild-type or other mutant worms. Does this suggest daf-9 and car-1 is partially overlapping/interdependent? (Compare Figure 3—figure supplement 1A and B and Figure 2—figure supplement 1A).

6) If CAR-1 drives lifespan extension by inhibiting GLP-1, why does a CAR-1 (WT) overexpression, which has a dramatic effect on number of germ cells have no effect on lifespan, and the very modest change in germ cells of CAR-1 (WT) overexpression versus CAR-1 K185R overexpression (this is a very modest decline compared to the dramatic decline versus wild-type) have such a profound effect on lifespan? (Compare Figure 3C-D to Figure 2—figure supplement 4A/C).

7) What was the purpose of measuring the effect of car-1 RNAi on egg laying of kri-1 mutant worms? It seems unsurprising that car-1 RNAi and kri-1 mutants, both of which decrease egg laying, have an additive phenotype. Moreover, the initial car-1 mutant experiments expressed that ¬car-1 mutant animals are sterile, so why would it be surprising that car-1 RNAi decreases egg-laying? Perhaps it would be less confusing if this section is removed, as it doesn't seem to add any additional benefits to the story. Alternatively, it can be moved into supplements as a control for the overexpression data (Figure 3E-F).

8) Is it surprising that EHC117 worms have no significant decrease in egg-laying when they have such a dramatic decline in germ cells? This should be textually addressed. This sentence can be added in the text where the differences in car-1 RNAi gonad versus egg-laying is compared – subsection “The roles of CAR-1 in GLP-1-mediated functions” (Compare Figure 3C-D to Figure 3I).

9) How do we know that the decreased egg-laying of CB1370 and car-1 RNAi are not just additive since both decrease egg-laying in otherwise WT animals? (Figure 3—figure supplement 2Figure)

10) Can you show that car-1 RNAi on its own does not affect sygl-1 activity? Why is this only done in the daf-2 mutant and not in WT animals? The authors should either include these controls, or explicitly state in the text why these experiments are not significant. The reviewers think that showing that overexpression of CAR-1 (WT or K185R) can decrease sygl-1 transcripts shows more direct evidence that CAR-1 negatively regulates transcriptional activity of GLP-1 (Figure 3J). Another suggestion is to stain for GLP-1 instead of measuring sygl-1 levels. K185R mutants should have decreased GLP-1 levels and car-1 RNAi should have decreased GLP-1 levels based on the model.

11) If car-1's effects are through GLP-1, car-1 RNAi should lead to an increase in GLP-1 protein. This should yield glp-1 gain of function phenotypes (e.g. persistent mitosis, increase in mitotic cells and fewer meiotic cells). Conversely, CAR-1 K185R should yield larger meiotic germ line cells causing a premature entry into meiosis (e.g. Pepper et al., 2003; Maine and Kimble, 1993). The absence of the phenotype in car-1 RNAi animals could suggests alternative mechanisms.

12) To further support the hypothesis that CAR-1 K185R that cannot be SUMOylated acts to increase longevity by decreasing GLP-1 would be to examine whether animals harboring CAR-1 K185R suppress glp-1 gain of function phenotypes (e.g. Pepper et al., 2003; Maine and Kimble, 1993), by sequestering glp-1 mRNA and decreasing its translation into protein.

13) Is the daf-16 CAR-1 K185 overexpression similar to daf-16 RNAi? This seems like an extremely important control to show that the two are not simply an additive effect if daf-16 RNAi has higher proteotoxicity. (Figure 4—figure supplement 1A). It would also be stronger argument that CAR-1 activity is falling within the daf-2 cascade if daf-2 RNAi/mutants do not have an additive effect with CAR-1 K185.

14) Why are all the paralysis experiments done with car-1 RNAi, which is an indirect way to show that car-1 is affecting proteoxicity through GLP-1? All of these experiments should be done with the CAR-1 K185 to show directly that hyperactive CAR-1 can actually protect against proteoxicity. Minimally, the text should explain why these experiments are done with RNAi to increase proteotoxicity (Figure 5 and Figure 5—figure supplement 1, Figure 5—figure supplement 3A-B).

15) For Figure 6A-C, why was CF512 worms used? Were these grown at 15, 20, or 25 C? Growth at 25 C can mildly stress worms and can affect HSR or thermotolerance. The actual growth conditions should be specified, and experiments should be repeated if worms were developed at 25 C. Moreover, the thermotolerance and *Pseudomonas* data seems to suggest that hsf-1 may be activated. It is recommended that the role of HSF-1 in these paradigms be tested.

16) For Figure 6D, if car-1 RNAi normally decreases lifespan, then the mild increase in lifespan is actually pretty substantial if normalized for this fact. Maybe something on this should be added in the text.

17) The experiments in Figure 6 really spark the curiosity of whether all of these phenotypes are truly due to car-1 or through non-specific pleotropic effects of car-1 RNAi. All of these experiments should be performed with the CAR-1 K185 strain to directly test whether these phenotypes are truly due to CAR-1. Moreover, would CAR-1 overexpression make worms more sensitive to pathogenic bacteria or UV (opposite of car-1 RNAi data), despite extending lifespan? Seems counter to the entire argument of the paper. These issues need to be addressed.

18) The article conflates fundamentally distinct concepts/roles, and in two cases that impact the overall conclusions. One, the role of Car1 in reproduction vs its role in longevity are considered interchangeably. The authors use effects on fertility to draw conclusions about longevity functions and vice versa and it is incorrect and confusing. Two, the function of GLP1 protein in germline development is substituted with the glp1 temperature sensitive mutant phenotype that is a surrogate for longevity brought on by germline loss. They use some results to directly implicate GLP1 protein function (e.g. sygl-1 expression) and others to deduce roles in germline eliminated longevity pathway (kri1, daf12 tests). This is also misleading and makes the article difficult to follow.

Car1's genetic interaction with Glp1 protein is published so if the premise is that it acts with/through Glp1 to modulate germline events that impact longevity, the experiments conducted here (Figure 3) do not test it or offer evidence for or against it. That requires examining the effect of CAR1 modulation on GLP1 (and the very well-characterized Notch pathway for translation control in the germline) within the germline and the fertility and lifespan consequences thereof. Similarly, if the hypothesis is that the Car1-Glp1 relationship operates in the context of germline-less longevity paradigm, the evidence provided is insufficient to infer any conclusions. The daf16/kri1/daf12/daf36 pathway is activated upon germline loss and solely impacts somatic Daf16. The relationships of these proteins within the germline are different (indeed, even in germline development these relationships change based on developmental state and/or environmental conditions). The effects of Car1 manipulations on reproduction of kri1 or daf12 mutants do not reveal their relationship in determining longevity, especially of sterile glp1 mutants. The germline-less longevity pathway has been shown by numerous studies to be genetically parallel to IIS and the authors overlook this concept while concluding that Car1 integrates 'aging controlling functions of IIS and germline'.

19) An alternative hypothesis that has not been addressed and would be consistent with much of the data such as the observed effects of overexpressing both wild-type CAR-1 and CAR-1 K185R on the number of germ line nuclei, is the role of CAR-1 on germline apoptosis (Boag et al., 2005). Programmed cell death mutations in *C. elegans* affect stress resistance, albeit reports show that the mechanisms are rather complex (e.g. Judy et al., 2013). It would strengthen the manuscript if this were directly examined, but this point needs to be addressed, at the very least in the Discussion section.

---

## [Author Response]

Essential revisions:1) Is the increased sumoylation found in CF512 (Figure 1A) due to differences that result in growth at 25 °C? Were the N2 worms also grown at 25 °C for this assay to control for this difference? The Materials and methods do not make it clear – it looks like it was performed at different temperatures. Several studies have shown that 25 °C can cause dramatic differences in proteostasis, so this needs to be re-evaluated or made clearer in the text. The authors are also urged to check other reproduction-deficient mutants, such as glp-1 and glp-4, or via chemical sterilization, such as FUDR, to see if these major differences are due to the lack of germline, or specific to CF512. If it is specific to CF512 and the reasoning behind why there are such major differences in sumoylation cannot be rectified, this begs the question of whether this is an appropriate model to use. Another question that comes up is whether CAR-1 protein itself is differentially sumoylated in CF512 background.

We agree with the referees that a clarification of methods and text were needed here. To clarify this issue, we first reproduced the experiment that is depicted as Figure 1A and modified the text to better explain the rationale. In the new experiment, we added a third worm strain that lack functional *glp-1* when exposed to 25ᵒC during development (strain CF1903). Similarly, to CF512 worms, the exposure of CF1903 worms to 25ᵒC renders the animals sterile. The experiment was conducted according to a revised protocol in which all worm strains were cultured in identical temperatures. All three parental (F0) worm populations (N2, CF512, and CF1903) were grown at 15ᵒC to ensure fertility, eggs were extracted and placed on plates that were seeded with either control bacteria (EV) or on *daf-2* RNAi bacteria. All plates were incubated for 48 hours at 25ᵒC and transferred to 20ᵒC for additional 24 hours, until all worms completed their development and reached day 1 of adulthood. As expected, CF512 and CF1903 worm populations, but not N2 animals, were sterile as displayed in Figure 1—figure supplement 1 of the revised manuscript. The worms were homogenized and total proteins were separated on a 10% PAA gel and blotted with a SUMO antibody.

We have modified the text (subsection “IIS reduction results in differential protein SUMOylation in *C. elegans*”) and the Materials and methods section to accurately explain how the experiment was performed. We thank the reviewers for this important comment.

2) Figure 1C-D are confusing. How do we know that 1C is showing SUMOylated-GFP-CAR-1 and not other protein that gets pulled down with an anti-SUMO antibody? Is this actually a double-IP where it is pulled down with SUMO antibodies, then GFP antibodies? This is especially confusing if probing with GFP-antibodies cannot visualize the larger, SUMOylated versions of the proteins – does SUMOylation prevent the GFP antibody from binding? Please clear up the experimental details and the conclusions drawn from the data.

We agree with the referees, revised the text and updated the Figure (1C) to better explain the experimental procedure and prevent confusion (please see subsection “IIS reduction lessens the SUMOylation of CAR-1”). In brief, WH346 worms were developed from hatching on either EV or daf-2 RNAi bacteria. At day 1 of adulthood, the worms (~300,000 per treatment) were homogenized and GFP-CAR-1 was pulled down by a GFP antibody. The sediment proteins were separated on a gel and blotted with a SUMO antibody. The large number of worms was needed due to the relative weak signal of SUMOylated proteins. In Figure 1D we display the same pulldown experiment as in 1C after reblotting it with a GFP antibody. Thus, Figure 1D serves as a loading control for Figure 1C. Since the signal of GFP antibody is much stronger than that of the SUMO antibody, the short exposure of the blot was not sufficient to detect SUMOylated proteins. To better analyze the total amounts of GFP-CAR-1 in these worms, we added Figure 1E, which displays another experiment in which we used a smaller number of worms (~4000 per treatment) and whole worm homogenates were loaded onto the gel. This result supports the conclusion that the knockdown of *daf-2* does not affect the levels of GFP-CAR-1. In addition, we labelled Figure 1C-E to indicate what is displayed in each panel, modified the text to clearly explain these experiments and changed the figure legend accordingly.

3) Since car-1 RNAi/KO decreases lifespan of wild-type and daf-2 mutants to a similar extent, is this just an additive phenotype? (Figure 2A-B.)

This is a valid point. We agree with the referees, tested how the knockdown of *car-1* by RNAi affects the lifespan of an additional *daf-2* mutant strain (*e1368*, Figure 2—figure supplement 1B), and modified the text to explain this issue. We added the sentence: “Yet, the lifespan reduction that we observed among untreated and *car-1* RNAi-treated *daf-2 (e1370*) mutant worms, which was similar to the difference observed among wild-type and *car-1* knockout animals (Figure 2A), questioned the notion that CAR-1 is involved in IIS-mediated regulation of lifespan” to subsection “The roles of CAR-1 in the regulation of lifespan”, to reflect this issue. We also suggest that various longevity mechanisms are regulated by the insulin/IGF signaling cascade, and thus, the SUMOylation of CAR-1 is only one such IIS-controlled mechanism. Accordingly, the effect of *car-1* RNAi on the lifespan of *daf-2* mutant worms is partial but may be related to the activity of this pathway. Please also see the Discussion section of the revised manuscript.

We thank the referees for highlighting this important issue.

4) A single copy rescue experiment of the CAR-1 K185R should be performed in the car-1 KO to ensure that this CAR-1 still maintains the normal (albeit hyperactive) functions of the protein (e.g. rescues sterility) and isn't a completely mutant form of the protein with brand new functions.

This is an important point that we tried addressing by injecting *car-1* heterozygous worms (strain WH377) with the CAR-1 K185 construct and searched for fertile worms that lack the endogenous *car-1* and express the CAR-1 K185R protein. Unfortunately, we could not identify such worms. This was plausibly due to the impaired fertility of worms that over-express the mutated *car-1* K185R (Figure 3G). Although we could not solve this issue experimentally within the time devoted for revision, we mentioned this important issue in the Discussion section of the revised manuscript: “(it is important to note that it is not clear whether this mutated CAR-1 restores the natural functions of the protein or activate another lifespan extending mechanism)”.

5) The lifespan decrease of car-1 RNAi on daf-9 mutants seems much milder than in wild-type or other mutant worms. Does this suggest daf-9 and car-1 is partially overlapping/interdependent? (Compare Figure 3—figure supplement 1A and B and Figure 2—figure supplement 1A).

We kindly disagree with the reviewers here, as the average differences in mean lifespans are similar. Each lifespan experiment was conducted three independent times. When *daf-2 (e1370*) mutant worms were fed with *car-1* RNAi the average shortening in mean lifespan following *car-1* RNAi treatment was 12.53% and *car-1* KO animals exhibited 15.32% lifespan shortening compared to their wild-type counterparts (see Supplementary file 2). Similarly, the knockdown of *car-1* by RNAi shortened the mean lifespan of *daf-9* mutant worms (CF2531) in 10.47% and of *daf-12* mutant animals (strain AA86) in 12.31% (see Supplementary file 4 for details). These differences are not significant and thus, we think that the shortening in mean lifespans are comparable and do not suggest an overlap in the pathways’ activities.

To better clarify this, we modified the text (please see subsection “The mechanisms of CAR-1-mediated lifespan regulation”).

6) If CAR-1 drives lifespan extension by inhibiting GLP-1, why does a CAR-1 (WT) overexpression, which has a dramatic effect on number of germ cells have no effect on lifespan, and the very modest change in germ cells of CAR-1 (WT) overexpression versus CAR-1 K185R overexpression (this is a very modest decline compared to the dramatic decline versus wild-type) have such a profound effect on lifespan? (Compare Figure 3C-D to Figure 2—figure supplement 4A/C).

We totally agree that this is an important issue, which requires clarification. It is likely that the over-expression of wild-type CAR-1 has only a little effect on lifespan, as the IIS promotes an efficient SUMOylation of the exogenous wtCAR-1 molecules and prevents them from affecting lifespan (Figure 2—figure supplement 4C). This may be similar to the case of DAF-16, which is efficiently phosphorylated by the IIS-regulated kinases and thus, *daf-*16 RNAi treatment has a small effect on the lifespan of wild-type worms. Since the CAR-1 K185R protein is SUMOylation-resistant, this mutant is not (or less) affected by the IIS and thus, the over-expression of this construct efficiently suppresses the activity of GLP-1 and extend lifespan (Figure 2D and E). Nevertheless, as noted accurately by the referees, the remarkable reducing effect of over-expressing wtCAR-1 on the number of germ cells is interesting and surprising (Figure 3D). This observation suggests that germ cells are more sensitive to perturbations in the levels of CAR-1 than lifespan. Accordingly, it is possible that although most exogenous CAR-1 molecules are SUMOylated by the IIS, a residual amount of non-SUMOylated CAR-1 molecules, could be sufficient to reduce the number of germ cells but not to extend lifespan. Importantly, the over-expression of the SUMOylation-resistant CAR-1 K185R further reduces the number of germ cells (Figure 3D). We further discussed this in subsection “The roles of CAR-1 in GLP-1-mediated functions” of the revised manuscript.

To further address the question why the over-expression of the wtCAR-1 reduces the number of germ cells (Figure 3D) but have a limited effect on the number of progeny (Figure 3H) we conducted an additional experiment. One possible explanation to this puzzling observation suggests that the over-expression of wtCAR-1 does not affect the number of apoptotic nuclei while expressing the mutated CAR-1 K185R does. If this is correct, the enhanced apoptosis may explain the different effects of the brood sizes. We tested this by comparing the number of apoptotic nuclei and found that indeed, the knockdown of *car-1* by RNAi, or the over-expression of the SUMOylation resistant CAR-1 K185R, but not of the wtCAR-1, elevate the number of apoptotic nuclei (Figure 3E). This new experiment is described in subsection “The roles of CAR-1 in GLP-1-mediated functions” of the revised manuscript.

7) What was the purpose of measuring the effect of car-1 RNAi on egg laying of kri-1 mutant worms? It seems unsurprising that car-1 RNAi and kri-1 mutants, both of which decrease egg laying, have an additive phenotype. Moreover, the initial car-1 mutant experiments expressed that ¬car-1 mutant animals are sterile, so why would it be surprising that car-1 RNAi decreases egg-laying? Perhaps it would be less confusing if this section is removed, as it doesn't seem to add any additional benefits to the story. Alternatively, it can be moved into supplements as a control for the overexpression data (Figure 3E-F).

The rationale behind this experiment was to examine whether the knockdown of *kri-1* and *car-1* have an additive effect. Yet, we agree with the referees that this rationale is not easy to follow and thus, we moved these results to the supplemental section as suggested. This experiment is now depicted as Figure 3—figure supplement 2C and D of the revised manuscript.

8) Is it surprising that EHC117 worms have no significant decrease in egg-laying when they have such a dramatic decline in germ cells? This should be textually addressed. This sentence can be added in the text where the differences in car-1 RNAi gonad versus egg-laying is compared – subsection “The roles of CAR-1 in GLP-1-mediated functions” (Compare Figure 3C-D to Figure 3I).

We thank the reviewers for this important suggestion and followed their guidance. As explained above, in the revised manuscript we also compared the number of apoptotic nuclei in the gonads of wild-type, EHC117 and EHC118 worms and found that the expression of the SUMOylation resistant CAR-1 K185R mutant, but not of the wtCAR-1 protein, elevates the numbers of apoptotic nuclei. This apparent contradiction between the number of germ cells and the brood size of worms that over-express the wtCAR-1 as well as the new experiment are described in subsection “The roles of CAR-1 in GLP-1-mediated functions” of the revised manuscript.

9) How do we know that the decreased egg-laying of CB1370 and car-1 RNAi are not just additive since both decrease egg-laying in otherwise WT animals? (Figure 3—figure supplement 2).

We agree that these effects could be additive and modified the text to reflect this issue. The sentence: “Yet, this reduction in brood size may be partially due to additive effects of IIS reduction and knocking down *car-1*” was added to subsection “The roles of CAR-1 in GLP-1-mediated functions”.

10) Can you show that car-1 RNAi on its own does not affect sygl-1 activity? Why is this only done in the daf-2 mutant and not in WT animals? The authors should either include these controls, or explicitly state in the text why these experiments are not significant. The reviewers think that showing that overexpression of CAR-1 (WT or K185R) can decrease sygl-1 transcripts shows more direct evidence that CAR-1 negatively regulates transcriptional activity of GLP-1 (Figure 3J). Another suggestion is to stain for GLP-1 instead of measuring sygl-1 levels. K185R mutants should have decreased GLP-1 levels and car-1 RNAi should have decreased GLP-1 levels based on the model.

This is an important point. We used here *daf-2* mutant worms as we expect that in these animals CAR-1 is much less SUMOylated (due to the low IIS activity), thereby should be more active. Thus, the effect of *car-1* RNAi is expected to be more prominent than in wild-type animals in which CAR-1 activity is suppressed by SUMOylation. However, we agree with the referee that a control experiment showing the effect of *car-1* RNAi on the levels of *sygl-1* in wild-type worms, is critical here. Accordingly, we used qPCR using N2 worms and *sygl-1* primers (Figure 3—figure supplement 4 of the revised manuscript). The results of five independent experiments show that, as expected, the knockdown of *car-1* by RNAi elevates the levels of *sygl-1* in N2 worms, however, this effect is not significant. This observation suggests that in wild-type worms CAR-1 is less active than in *daf-*2 mutant animals and thus, the effect of knocking down this gene on the expression of *sygl-1* is smaller than in *daf-2* mutant animals. This experiment is delineated in subsection “The levels of car-1 modulate the transcriptional activity of the GLP-1 pathway”.

The reviewers think that showing that overexpression of CAR-1 (WT or K185R) can decrease sygl-1 transcripts shows more direct evidence that CAR-1 negatively regulates transcriptional activity of GLP-1 (Figure 3J). Another suggestion is to stain for GLP-1 instead of measuring sygl-1 levels. K185R mutants should have decreased GLP-1 levels and car-1 RNAi should have decreased GLP-1 levels based on the model.

We addressed this important issue by measuring the expression levels of an additional GLP-1 target gene, *lst-1.* While the exposure of CF1903 worms to 25ᵒC remarkably reduced the expression levels of *lst-1* (Figure 3—figure supplement 3B), this gene showed reduced expression upon the knockdown of *car-1.* This shows that the relations between CAR-1 and GLP-1 are complex and probably depends on additional factors. We modified the text to discuss this important issue and thank the referees for their scientific guidance. Please see subsection “The levels of car-1 modulate the transcriptional activity of the GLP-1 pathway” and the Discussion section.

11) If car-1's effects are through GLP-1, car-1 RNAi should lead to an increase in GLP-1 protein. This should yield glp-1 gain of function phenotypes (e.g. persistent mitosis, increase in mitotic cells and fewer meiotic cells). Conversely, CAR-1 K185R should yield larger meiotic germ line cells causing a premature entry into meiosis (e.g. Pepper et al., 2003; Maine and Kimble, 1993). The absence of the phenotype in car-1 RNAi animals could suggests alternative mechanisms.

We took two measures to expand our analysis of the relations between CAR-1 and GLP-1. First, we compared the number of apoptotic nuclei in the gonads of wild type, EHC117 and EHC118 worms. In addition, we followed the expression levels of an additional GLP-1 target gene, *lst-1.* The comparison of apoptotic nuclei in the gonads of EHC117, EHC118 and wild-type animals is now displayed as Figure 3E and discussed in subsection “The roles of CAR-1 in GLP-1-mediated functions”. The complex relations of CAR-1 and GLP-1 are demonstrated by the opposing effects of *car-1* RNAi on the expression levels of sygl-1 and lst-1 and explained in subsection “The levels of car-1 modulate the transcriptional activity of the GLP-1 pathway” and the Discussion section.

12) To further support the hypothesis that CAR-1 K185R that cannot be SUMOylated acts to increase longevity by decreasing GLP-1 would be to examine whether animals harboring CAR-1 K185R suppress glp-1 gain of function phenotypes (e.g. Pepper et al., 2003; Maine and Kimble, 1993), by sequestering glp-1 mRNA and decreasing its translation into protein.

This is an important comment. We tried to adopt a functional approach and injected GC833 worms ((*glp-1*(ar202) III), that develop tumor due to the activation of GLP-1 (Pepper et al., Genetics 2003)) with the CAR-1 K185R plasmid. However, this rendered the animals sterile and we could not isolate rescued animals.

Thus, we modified the text to explain the complex relations between CAR-1 and GLP-1 (Discussion section).

13) Is the daf-16 CAR-1 K185 overexpression similar to daf-16 RNAi? This seems like an extremely important control to show that the two are not simply an additive effect if daf-16 RNAi has higher proteotoxicity. (Figure 4—figure supplement 1A). It would also be stronger argument that CAR-1 activity is falling within the daf-2 cascade if daf-2 RNAi/mutants do not have an additive effect with CAR-1 K185.

We thank the reviewers for this important comment. We modified the text to add a comparison of the rates of paralysis in *daf-16* RNAi-treated CL2006 worms in published articles (Cohen et al., 2006, Cohen et al., 2010) to the rate seen in EHC124 animals that were grown on *daf-16* RNAi bacteria (Figure 4—figure supplement 1A). The similar rates of paralysis suggest that the protective effect of CAR-1 K185R is *daf-16* dependent. This is described in subsection “CAR-1 is involved in maintaining proteostasis”.

14) Why are all the paralysis experiments done with car-1 RNAi, which is an indirect way to show that car-1 is affecting proteoxicity through GLP-1? All of these experiments should be done with the CAR-1 K185 to show directly that hyperactive CAR-1 can actually protect against proteoxicity. Minimally, the text should explain why these experiments are done with RNAi to increase proteotoxicity (Figure 5 and Figure 5—figure supplement 1, Figure 5—figure supplement 3A-B).

In Figure 4 we tested whether CAR-1 is a modulator of proteotoxicity. To address this we used: (i) *car-1* RNAi (Figure 4A and B), (ii) mixes of *car-1* RNAi with other RNAi bacterial strains (Figure 4C and D) and (iii) Aβ worms which express the CAR-1 K185R proteins (Figure 4E and F). We also tested whether the protection that is conferred by CAR-1 K185R is *daf-16* dependent and whether the CAR-1 K257R affects Aβ toxicity (Figure 4—figure supplement 1A and B). The results of all of these experiments culminate to show that the knockdown of *car-1* expose the worms to Aβ proteotoxicity, the over-expression of the hyper-active CAR-1 K185R, but not of the CAR-1 K257R, protects the animals from Aβ toxicity in a *daf-16* dependent manner.

The rationale behind the experiments that we display in Figure 5A and B, Figure 5—figure supplement 1 and Figure 5—figure supplement 3 is different. Here we addressed two questions. First, we asked whether CAR-1 also modulates proteotoxicity of an additional disease-linked, aggregation-prone protein, polyQ-YFP. We found that similarly to the effect seen in Aβ worms, the knockdown of *car-1* exposes the worms to proteotoxicity in a *glp-1* dependent manner. Secondly, we asked whether the RNA helicase CGH-1, is also a modulator of proteotoxicity. Our results (Figure 5C-D and Figure 5—figure supplement 3) show that it indeed modulates proteotoxicity.

We modified the text as suggested, to better explain the experiments of Figure 5 and Figure 5—figure supplement 3 (Results section).

15) For Figure 6A-C, why was CF512 worms used? Were these grown at 15, 20, or 25 C? Growth at 25 C can mildly stress worms and can affect HSR or thermotolerance. The actual growth conditions should be specified, and experiments should be repeated if worms were developed at 25 C. Moreover, the thermotolerance and Pseudomonas data seems to suggest that hsf-1 may be activated. It is recommended that the role of HSF-1 in these paradigms be tested.

We agree with the referees and repeated the heat tolerance experiment that is presented as Figure 6A using N2 worms. Our new results (Figure 6B) show that similarly to the observations that we obtained using CF512 worms, the knockdown of *car-1* by RNAi has no significant effect on heat tolerance of wild-type worms. In addition, in our previous study (Volovik et al., 2012) we have shown no HSF-1 activation in CF512 animals that were developed in 25ᵒC and transferred to 20ᵒC at day 1 of adulthood (as judged by the lack of induction of the HSF-1-target gene *hsp-16.2*). To clarify this issue we cited our 2012 Aging Cell paper and to describe the new experiment (Results section).

16) For Figure 6D, if car-1 RNAi normally decreases lifespan, then the mild increase in lifespan is actually pretty substantial if normalized for this fact. Maybe something on this should be added in the text.

We agree with the referees, conducted additional experiments and modified the text to better explain the slight increase in resistance to *Pseudomonas aeruginosa.* First, we examined whether the over-expression of wtCAR-1 (strain EHC117) and/or of CAR-1 K185R (strain EHC118) affect the survival of worms that were exposed to *Pseudomonas aeruginosa.* We found that the over-expression of *car-1* (wild-type or mutant) slightly shortens the worms’ survival. These results (Figure 6F) support the notion that CAR-1 activity is deleterious when the animals are exposed to pathogenic bacteria. We also expanded the textual explanation to better discuss this important issue (please see Results section).

17) The experiments in Figure 6 really spark the curiosity of whether all of these phenotypes are truly due to car-1 or through non-specific pleotropic effects of car-1 RNAi. All of these experiments should be performed with the CAR-1 K185 strain to directly test whether these phenotypes are truly due to CAR-1. Moreover, would CAR-1 overexpression make worms more sensitive to pathogenic bacteria or UV (opposite of car-1 RNAi data), despite extending lifespan? Seems counter to the entire argument of the paper. These issues need to be addressed.

We thank the referees for highlighting this important issue and conducted the suggested experiment. We compared the survival of N2, EH117, and EHC118 worms (over-expressing the wild-type or K185R car-1 mutant respectively) that were exposed to heat (Figure 6D). We also tested whether EHC118 worms are more sensitive to pathogenic bacteria (Figure 6F) and to sub-lethal dose of UV radiation (Figure 6H). Our results show that, as predicted by the referees, the over-expression of the CAR-1 K185R acts in opposite to the knockdown of *car-1* by RNAi as it lowers survival of stressed animals.

These new experiments are described in the Results section of the revised manuscript.

It is important to note that lifespan, stress resistance and proteostasis were shown by multiple studies to be separable. For instance, the ability to respond to heat stress comes at the expense of the worms capacity to cope with proteotoxicity (Prahlad and Morimoto, 2011). Similarly, we have shown (Maman et al., 2013) that the knockdown of *gtr-*1 elevates heat sensitivity but has no effect on lifespan. Thus, in the light of these insights, the opposing effects of *car-1* RNAi on lifespan and stress resistance are not surprising.

We further discussed the relations between lifespan and stress resistance in the revised Discussion section.

18) The article conflates fundamentally distinct concepts/roles, and in two cases that impact the overall conclusions. One, the role of Car1 in reproduction vs its role in longevity are considered interchangeably. The authors use effects on fertility to draw conclusions about longevity functions and vice versa and it is incorrect and confusing. Two, the function of GLP1 protein in germline development is substituted with the glp1 temperature sensitive mutant phenotype that is a surrogate for longevity brought on by germline loss. They use some results to directly implicate GLP1 protein function (eg., sygl-1 expression) and others to deduce roles in germline eliminated longevity pathway (kri1, daf12 tests). This is also misleading and makes the article difficult to follow.

We see the point here and took several measures to clarify the manuscript and avoid confusion.

First, since theoretically it possible that, in addition to the reported link between CAR-1 and GLP-1 (Noble et al., 2008), CAR-1 influences additional cellular pathways, we sought to directly test if the knockdown of *car-1* affects the transcriptional activity of the GLP-1-controlled pathway. To address this, we tested whether the knockdown of *car-*1 by RNAi affects the expression of *sygl-1* and *lst-1*, both well-defined targets of the GLP-1 pathway. Our results (Figure 3I) provide a direct evidence to the negative regulation of CAR-1 on the transcriptional activity of GLP-1 on *sygl-1* but not on *lst-1*. This highlights the complex relations of CAR-1 and GLP-1 that are now explained in the Results section and the Discussion section.

Secondly, to further examine the roles of CAR-1 of GLP-1 mediated functions, we tested how CAR-1 germ cell proliferation (Figure 3, C and D) and egg laying (Figure 3, F-H). In addition, since GLP-1 regulates the activity of DAF-16 through a well-defined set of components, including KRI-1 (Berman and Kenyon, 2006), we tested whether *car-1* RNAi affects egg laying patterns and lifespan of worms that lack functional *kri-1.* We found that it reduces the brood size of *kri-1* mutant worms (Figure 3—figure supplement 2C and D) but does not affect the lifespan of these animals (Figure 3B).

Finally, in the revised manuscript we tested how GLP-1 affects the expression level of *car-1* (Figure 3J) and found that the expression of *car-1* is lower in worms that lack functional GLP-1 compared to their counterparts the harbor functional GLP-1.

Together, these results describe the complex relations of CAR-1 and of GLP-1. The text has been modified to further discuss these relations (Results section).

Car1's genetic interaction with Glp1 protein is published so if the premise is that it acts with/through Glp1 to modulate germline events that impact longevity, the experiments conducted here (Figure 3) do not test it or offer evidence for or against it. That requires examining the effect of CAR1 modulation on GLP1 (and the very well-characterized Notch pathway for translation control in the germline) within the germline and the fertility and lifespan consequences thereof.

Our results show that the knockdown of *car-1* shortens lifespan of wild-type and *daf-2* mutant worms (Figure 2A-B) but not of worms that lack functional GLP-1 (Figure 3A). This shows that the effect of CAR-1 on lifespan is GLP-1 dependent. Moreover, the knockdown of *car-1* modulates the transcription of GLP-1-target genes (*sygl-1* and *lst-1*), showing that CAR-1 regulates at least some of the activities of GLP-1.

We also tested how the knockdown and over-expression of *car-1* affect the number of germ cells, reproduction profile, and number of apoptotic nuclei (Figures 3C-E and G-J).

We hope that the modified text (Results section) as well as the wealth of new results further clarify these issues raised here by the referees.

Similarly, if the hypothesis is that the Car1-Glp1 relationship operates in the context of germline-less longevity paradigm, the evidence provided is insufficient to infer any conclusions.

Our hypothesis suggests that since CAR-1 regulates the activity of GLP-1, as the CAR-1-GLP-1 relationship should have no effect on lifespan of worms that lack functional GLP-1 (Figure 3A) or functional DAF-16 (Figure 2C). In addition, the expression of CAR-1 K185R extends lifespan. These results clearly show that CAR-1 is a lifespan regulator, probably through the GLP-1 controlled mechanism.

The daf16/kri1/daf12/daf36 pathway is activated upon germline loss and solely impacts somatic Daf16. The relationships of these proteins within the germline are different (indeed, even in germline development these relationships change based on developmental state and/or environmental conditions). The effects of Car1 manipulations on reproduction of kri1 or daf12 mutants do not reveal their relationship in determining longevity, especially of sterile glp1 mutants. The germline-less longevity pathway has been shown by numerous studies to be genetically parallel to IIS and the authors overlook this concept while concluding that Car1 integrates 'aging controlling functions of IIS and germline'.

The key conclusion of this work is that indeed, the aging-regulating pathways, not necessarily longevity, but stress resistance and proteostasis, downstream of the IIS and the germ cells are inter-related through the SUMOylation of CAR-1. Our data challenge the conclusion of previous studies that claimed that these two pathways are independent. Nevertheless, we expanded the Discussion section to better discuss this paradigm.

19) An alternative hypothesis that has not been addressed and would be consistent with much of the data such as the observed effects of overexpressing both wild-type CAR-1 and CAR-1 K185R on the number of germ line nuclei, is the role of CAR-1 on germline apoptosis (Boag et al., 2005). Programmed cell death mutations in C. elegans affect stress resistance, albeit reports show that the mechanisms are rather complex (e.g. Judy et al., 2013). It would strengthen the manuscript if this were directly examined, but this point needs to be addressed, at the very least in the Discussion section.

We followed the referee’s guidance and tested the proposed hypothesis by comparing the number of apoptotic nuclei in the gonads of untreated and *car-1* RNAi-treated wild-type worms, as well as in EHC117 and EHC118 animals. Our results (Figure 3E of the revised manuscript) show that the knockdown of *car-*1 by RNAi, and the over-expression of the SUMOylation resistant CAR-1 K185R elevate the numbers of apoptotic nuclei in the gonad. No such effect was seen when the wtCAR-1 was over-expressed. These results provide a possible explanation to the lack of significant difference in the brood sizes of N2 and EHC117 worms (Figure 3H). We thank the referee for this important comment and agree that this mechanism is complex. Accordingly, we expanded the Discussion section to explain the roles of CAR-1 in controlling germ cells apoptosis.